# Higher Order Kernel Mean Embeddings to Capture Filtrations of Stochastic Processes

**Cristopher Salvi** [1]
cristopher.salvi@maths.ox.ac.uk

**Maud Lemercier** [2]
maud.lemercier@warwick.ac.uk

**Chong Liu** [3]
chong.liu@maths.ox.ac.uk

**Blanka Horvath** [4]
blanka.horvath@kcl.ac.uk

**Theodoros Damoulas** [2]
t.damoulas@warwick.ac.uk

**Terry Lyons** [1]
terry.lyons@maths.ox.ac.uk

## Abstract

Stochastic processes are random variables with values in some space of paths. However, reducing a stochastic process to a path-valued random variable ignores its *filtration*, i.e. the flow of information carried by the process through time. By conditioning the process on its filtration, we introduce a family of *higher order kernel mean embeddings* (KMEs) that generalizes the notion of KME and captures additional information related to the filtration. We derive empirical estimators for the associated *higher order maximum mean discrepancies* (MMDs) and prove consistency. We then construct a filtration-sensitive kernel two-sample test able to pick up information that gets missed by the standard MMD test. In addition, leveraging our higher order MMDs we construct a family of universal kernels on stochastic processes that allows to solve real-world calibration and optimal stopping problems in quantitative finance (such as the pricing of American options) via classical kernel-based regression methods. Finally, adapting existing tests for conditional independence to the case of stochastic processes, we design a causal-discovery algorithm to recover the causal graph of structural dependencies among interacting bodies solely from observations of their multidimensional trajectories.

## 1  Introduction

The idea of embedding probability distributions into a reproducing kernel Hilbert space (RKHS) via kernel mean embeddings (KMEs) has become ubiquitous in many areas of statistics and data science such as hypothesis testing [1, 2], non-linear regression [3, 4], distribution regression [5, 6] etc. Despite strong progress in the study of KMEs, most of the examples considered in the literature tend to focus on random variables supported on some finite (possibly high) dimensional euclidean spaces like $\mathbb{R}^d$. The study of KMEs for function-valued random variables has been largely ignored.

Stochastic processes are random variables with values in some space of paths. However, reducing a stochastic process to a path-valued random variable ignores its *filtration*, which can be informally thought of as the *flow of information carried by the process through time*. A question that naturally

---

[1]University of Oxford & The Alan Turing Institute

[2]University of Warwick & The Alan Turing Institute

[3]University of Oxford

[4]King's College London, The Alan Turing Institute & Technical University of Munich

35th Conference on Neural Information Processing Systems (NeurIPS 2021).

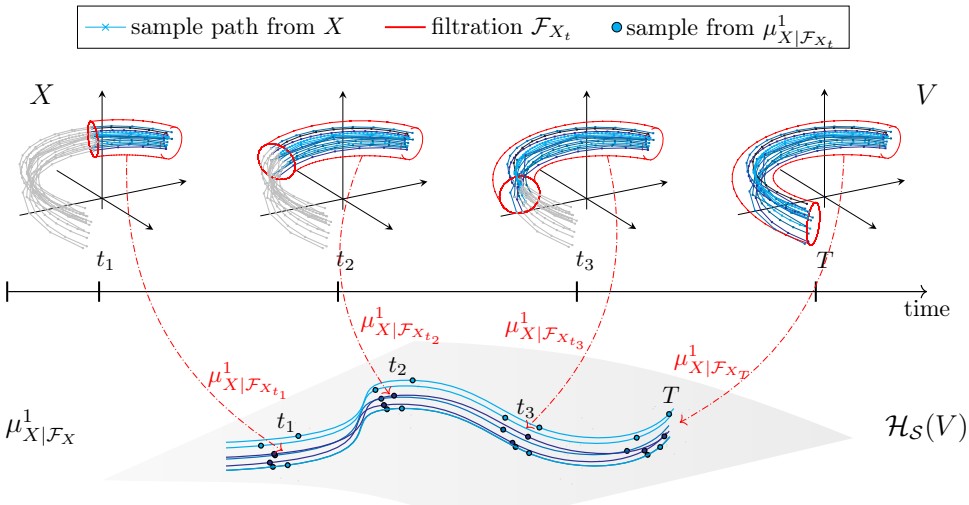

Figure 1: Schematic overview of the construction of the $1^{\text{st}}$ order predictive KME $\mu^1_{X|\mathcal{F}_X}$ (Sec. 3.2). Here $X$ is a stochastic process with sample paths taking their values in $V$. The red contours indicate the portion of its filtration $\mathcal{F}_X$ upon which the conditioning is applied, i.e. the available information about $X$ from start up to time $t$. As explained in Sec. 3.2, the $1^{\text{st}}$ order predictive KME $\mu^1_{X|\mathcal{F}_X}$ is a path whose value at time $t$ is a $\mathcal{H}_\mathcal{S}(V)$-valued random variable representing the law of $X$ conditioned on its filtration $\mathcal{F}_{X_t}$. Equivalently $\mu^1_{X|\mathcal{F}_X}$ is a stochastic process with sample paths taking their values in $\mathcal{H}_\mathcal{S}(V)$.

emerges from the study of many random, time-evolving systems like financial markets is how does the information available up to present time affect the future evolution of the system?

Formally, this question can be addressed by conditioning a process on its filtration (Sec. 3.1 and 3.2). In this paper we introduce a family of *higher order KMEs* that generalizes the notion of KME to capture additional, filtration-related information (Sec. 3.3 and 3.5). In view of concrete applications, we derive empirical estimators for the associated *higher order MMDs* and use one of them to construct a filtration-sensitive kernel two-sample test (Sec. 3.4) demonstrating with simulated data its ability to capture information that otherwise gets missed by the standard MMD test (Sec. 4.1). Furthermore, we construct a family of universal kernels on stochastic processes (Sec. 3.6) that allows to solve challenging, real-world optimisation problems in quantitative finance such as the pricing of American options via classical kernel-based regression methods (Sec. 4.2). Finally, we adapt existing tests for conditional independence to the case of stochastic processes in order to design a causal-discovery algorithm able to recover the causal graph of structural dependencies among interacting bodies solely from observations of their multidimensional trajectories (Sec. 4.3).

## 1.1 Related work

The notion of conditioning is a powerful probabilistic tool allowing to understand possibly complex, non-linear interactions between random variables. As their unconditional counterparts, conditional distributions can also be embedded into RKHSs [7]. Recently, conditional KMEs have received increased attention, especially in the context of graphical models [8], state-space models [9], dynamical systems [10], causal inference [11, 12, 13], two-sample and conditional independence hypothesis testing [14, 13, 15] and others. Embeddings of distributions via KMEs have also shown their success in the context of distribution regression (DR), which is the task of learning a function mapping a collection of samples from a probability distribution to scalar targets [16, 17, 18]. More recently, a framework for DR that addresses the setting where inputs are sample paths from an underlying stochastic process is proposed in [6]. The authors make extensive use of the *signature transform* [19, 20] and of the *signature kernel* [21, 22], two well established tools in stochastic analysis.

When it comes to stochastic processes, it was first shown in [23] that weak convergence of random variables does not always account for the information contained in the filtration, as highlighted by means of numerous numerical examples in [24, 25]. This limitation is addressed in [23, 26] through

the construction of a sequence of so–called *adapted topologies*[5] $(\tau_n)_{n\geq 1}$ that become progressively finer[6] as $n$ increases (with $\tau_1$ coinciding with the weak topology). In particular, higher order adapted topologies are shown to capture more filtration-related information than their weak counterpart. This characteristic becomes relevant for example in some optimal stopping problems such as the pricing of American options, where the pricing function can be shown to be discontinuous with respect to the weak topology, but is continuous with respect to the second order adapted topology[7] [24, 28, 27]. Leveraging properties of the signature transform, it has been shown that adapted topologies are intimately related to a family of higher order MMDs [29]. However, providing empirical estimators for these discrepancies that can be deployed on real-world tasks remains a challenge. In this paper we propose to address this challenge by presenting an alternative construction to this higher order MMDs using the language of kernels and KMEs. The results in [29] serve as a strong theoretical background for the present paper.

## 2 Preliminaries

We begin with a brief summary of tools from stochastic analysis needed to define higher order KMEs. Let $\mathcal{X}(\mathbb{R}^d) = \{x : [0, T] \to \mathbb{R}^d\}$ a compact set of continuous, piecewise linear, $\mathbb{R}^d$-valued paths defined over a common time interval $[0, T]$, obtained for example by linearly interpolating a multivariate time series. For any path $x \in \mathcal{X}(\mathbb{R}^d)$ we denote its $k^{\text{th}}$ coordinate by $x^{(k)} : [0, T] \to \mathbb{R}$, for $k \in \{1, \ldots, d\}$. More generally we denote by $\mathcal{X}(V) = \{x : [0, T] \to V\}$ a compact set of continuous, piecewise linear paths with values in a Hilbert space $V$ with a countable basis.

### 2.1 The signature transform

The *signature transform* $\mathcal{S} : \mathcal{X}(V) \to H_{\mathcal{S}}(V)$ is a *feature map* defined for any path $x \in \mathcal{X}(V)$ as the following infinite collection of statistics

$$\mathcal{S}(x) = \left( 1, \left\{ \mathcal{S}(x)^{(k_1)} \right\}_{k_1=1}^{d}, \left\{ \mathcal{S}(x)^{(k_1,k_2)} \right\}_{k_1,k_2=1}^{d}, \ldots \right) \tag{1}$$

where each term is a real number equal to the iterated integral

$$\mathcal{S}(x)^{(k_1,\ldots,k_j)} = \int \ldots \int_{0 < s_1 < \ldots < s_j < T} dx_{s_1}^{(k_1)} \ldots dx_{s_j}^{(k_j)} \tag{2}$$

The signature *feature space* $H_{\mathcal{S}}(V)$ is defined as the following direct sum of tensor powers of $V$

$$H_{\mathcal{S}}(V) = \bigoplus_{k=0}^{\infty} V^{\otimes k} = \mathbb{R} \oplus V \oplus (V)^{\otimes 2} \oplus \ldots \tag{3}$$

where $\otimes$ denotes the standard tensor product of vector spaces [30, 19].

### 2.2 The signature kernel

Because $V$ is Hilbert $H_{\mathcal{S}}(V)$ is also Hilbert [21]. The *signature kernel* $k_{\mathcal{S}} : \mathcal{X}(V) \times \mathcal{X}(V) \to \mathbb{R}$ is a characteristic kernel defined for any pair of paths $x, y \in \mathcal{X}(V)$ as the following inner product

$$k_{\mathcal{S}}(x, y) = \langle \mathcal{S}(x), \mathcal{S}(y) \rangle_{H_{\mathcal{S}}(V)} \tag{4}$$

The recent article [22] establishes a surprising connection between the signature kernel and a certain class of partial differential equations (PDEs), culminating in the following kernel trick for $k_{\mathcal{S}}$.

**Theorem 1.** *[22, Thm. 2.5] For any $x, y \in \mathcal{X}(V)$ the signature kernel satisfies the equation $k_{\mathcal{S}}(x, y) = u_{x,y}(T, T)$, where $u_{x,y} : [0, T] \times [0, T] \to \mathbb{R}$ is the solution of the hyperbolic PDE*

$$\frac{\partial^2 u_{x,y}}{\partial s \partial t} = \langle \dot{x}_s, \dot{y}_t \rangle_V \, u_{x,y} \tag{5}$$

*with boundary conditions $u_{x,y}(0, \cdot) = u_{x,y}(\cdot, 0) = 1$ and where $\dot{z}_s = \frac{dz_r}{dr}\big|_{r=s}$.*

---

[5]We say that a sequence of random variables $\{X_n\}_{n\in\mathbb{N}}$ converges to a random variable $X$ in the topology $\tau$ if and only if for every $\tau$-open neighbourhood $\mathbb{U}$ of $\mathbf{X}$ there exists $N \in \mathbb{N}$ such that $X_n \in \mathbb{U}$ as soon as $n \geq N$.

[6]A topology $\tau_1$ is said to be finer than a topology $\tau_2$ if every $\tau_2$-open set is also $\tau_1$-open.

[7]The second order adapted topology $\tau_2$ is equivalent to the *adapted Wasserstein distance* [27].

Hence, evaluating $k_{\mathcal{S}}$ at a pair of paths $(x, y)$ is equivalent to solving the PDE (5); in this paper we solve PDEs numerically via a finite difference scheme (see Appendix B for additional details). In what follows, we denote by $\mathcal{H}_{\mathcal{S}}(V)$ the RKHS associated to $k_{\mathcal{S}}$.

## 2.3 Stochastic processes and filtrations

We take $(\Omega, \mathcal{F}, \mathbb{P})$ as the underlying probability space. A *(discrete time) stochastic process* $X$ is a random variable with values on $\mathcal{X}(V)$. We denote by $\mathbb{P}_X = \mathbb{P} \circ X^{-1}$ the *law* of $X$. Assuming the integrability condition $\mathbb{E}_X[k_{\mathcal{S}}(X, X)] < \infty$, the 1st *order kernel mean embedding* (KME) of $X$ is defined as[8] the following point in $\mathcal{H}_{\mathcal{S}}(V)$

$$\mu_X^1 = \mathbb{E}_X[k_{\mathcal{S}}(X, \cdot)] = \int_{x \in \mathcal{X}(V)} k_{\mathcal{S}}(x, \cdot) \mathbb{P}_X(dx) \tag{6}$$

Accordingly, given two stochastic processes $X, Y$, their 1st *order maximum mean discrepancy* (MMD) is the standard MMD distance with kernel $k_{\mathcal{S}}$ given by the following expression

$$\mathcal{D}_{\mathcal{S}}^1(X, Y) = \left\| \mu_X^1 - \mu_Y^1 \right\|_{\mathcal{H}_{\mathcal{S}}(V)} \tag{7}$$

Because the signature kernel $k_{\mathcal{S}}$ is characteristic, it is a classical result [1, 31] that the 1st order MMD is a sufficient statistics to distinguish between the laws of $X$ and $Y$, in other words

$$\mathcal{D}_{\mathcal{S}}^1(X, Y) = 0 \iff \mathbb{P}_X = \mathbb{P}_Y \tag{8}$$

Despite the fact that stochastic processes are path-valued random variables, they encode a much richer structure compared to standard $\mathbb{R}^d$-valued random variables, that goes well beyond their laws. This additional structure is described mathematically by the concept of *filtration* of a process $X$, defined as the following family of $\sigma$-algebras

$$\mathcal{F}_X = (\mathcal{F}_{X_t})_{t \in [0,T]}, \tag{9}$$

where for any $t \in [0, T]$, $\mathcal{F}_{X_t}$ is the $\sigma$-algebra generated by the variables $\{X_s\}_{s \in [0,t]}$. Note that $\mathcal{F}_X$ is totally ordered in the sense that $\mathcal{F}_{X_s} \subset \mathcal{F}_{X_t}$ for all $s < t$, which naturally explains why filtrations are good mathematical descriptions to model the flow information carried by the process $X$.

In the next section, we will present our main findings and introduce a family of *higher order KMEs* and corresponding *higher order MMDs* as generalizations of the standard KME and MMD respectively. We will do so by conditioning stochastic processes on elements of their filtrations.

# 3 Higher order kernel mean embeddings

We begin by describing how KMEs can be extended to conditional laws of stochastic processes.

## 3.1 Conditional kernel mean embeddings for stochastic processes

Let $X, Y$ be two stochastic processes. For a given path $x \in \mathcal{X}(V)$, define the 1st *order conditional kernel mean embeddings* $\mu_{Y|X=x}^1 \in \mathcal{H}_{\mathcal{S}}(V)$ and $\mu_{Y|X}^1 : \mathcal{H}_{\mathcal{S}}(V) \to \mathcal{H}_{\mathcal{S}}(V)$ as follows

$$\mu_{Y|X=x}^1 = \mathbb{E}[k_{\mathcal{S}}(Y, \cdot)|X = x] = \int_{y \in \mathcal{X}(V)} k_{\mathcal{S}}(\cdot, y) \mathbb{P}_{Y|X=x}(dy) \tag{10}$$

$$\mu_{Y|X}^1 = \mathbb{E}[k_{\mathcal{S}}(Y, \cdot)|X] = \int_{y \in \mathcal{X}(V)} k_{\mathcal{S}}(\cdot, y) \mathbb{P}_{Y|X}(dy) \tag{11}$$

Note that whilst $\mu_{Y|X=x}^1$ is a single point in $\mathcal{H}_{\mathcal{S}}(V)$, the 1st order conditional KME $\mu_{Y|X}^1$ describes a cloud of points on $\mathcal{H}_{\mathcal{S}}(V)$. Each point in this cloud is indexed by a path $x \in \mathcal{X}(V)$. Equivalently, $\mu_{Y|X}^1$ constitutes a $\mathbb{P}_X$-measurable, $\mathcal{H}_{\mathcal{S}}(V)$-valued random variable.

These embeddings allow to extend the applications of conditional KMEs to the case where the random variables are (possibly multidimensional) stochastic processes. In particular one can directly obtain conditional independence criterions for stochastic processes (see Appendix A.1), enabling to deploy standard kernel-based causal learning algorithms [13], as we demonstrate in Sec. 4. Next we describe how in the case of stochastic processes, conditioning on filtrations is an important mathematical operation to model real-world time-evolving systems.

---

[8]The 1st order KME is the standard KME with the signature kernel $k_{\mathcal{S}}$.

## 3.2 Conditioning stochastic processes on their filtrations

Financial markets are examples of complex dynamical systems that evolve under the influence of randomness. An important objective for financial practitioners is to determine how actionable information available up to present could affect the future market trajectories. The task of *conditioning on the past to describe the future* of a stochastic process $X$ can be formulated mathematically by conditioning $X$ on its filtration $\mathcal{F}_{X_t}$ for any time $t \in [0, T]$.

More precisely, consider the $1^{\text{st}}$ order KME of the conditional law $\mathbb{P}_{X|\mathcal{F}_{X_t}}$, which is defined as the following $\mathcal{F}_{X_t}$-measurable, $\mathcal{H}_\mathcal{S}(V)$-valued random variable

$$\mu^1_{X|\mathcal{F}_{X_t}} = \mathbb{E}[k_\mathcal{S}(X, \cdot)|X_{[0,t]}] = \int_{x \in \mathcal{X}(V)} k_\mathcal{S}(\cdot, x)\mathbb{P}_{X|\mathcal{F}_{X_t}}(dx) \tag{12}$$

where $X_{[0,t]}$ denotes the stochastic process $X$ restricted to the sub-interval $[0, t] \subset [0, T]$. By varying the time index $t$, we can form the following ordered collection of $1^{\text{st}}$ order KMEs

$$\mu^1_{X|\mathcal{F}_X} = \left(\mu^1_{X|\mathcal{F}_{X_t}}\right)_{t \in [0,T]} \tag{13}$$

that we term $1^{st}$ *order predictive KME* of the process $X$. By construction, $\mu^1_{X|\mathcal{F}_X}$ describes a path taking its values in the space of $\mathcal{H}_\mathcal{S}(V)$-valued random variables, in other words it is itself a stochastic process [9] (see Fig. 1). Hence, the law of $\mu^1_{X|\mathcal{F}_X}$ can itself be embedded via KMEs into a "higher-order RKHS" (see next section), making the full procedure iterable, as we shall discuss next.

We note that for each time $t$, the random variable $\mu^1_{X|\mathcal{F}_{X_t}}$ in eq. (12) is the Bochner integral of $k_\mathcal{S}(\cdot, x)$ with respect to the probability measure $\mathbb{P}_{X|\mathcal{F}_{X_t}}$. Since we assumed that $V$ is a compact set, the path space $\mathcal{X}(V)$ is also compact. Hence, the function $x \mapsto k_\mathcal{S}(\cdot, x)$ is continuous, the set $K = \{k_\mathcal{S}(\cdot, x) : x \in \mathcal{X}(V)\}$ is compact as continuous image of a compact set, and therefore its Bochner integral $\mu^1_{X|\mathcal{F}_{X_t}}$ takes values in the closed convex hull of $K$, which is again a compact subset in the RKHS $\mathcal{H}_\mathcal{S}(V)$. Consequently the path $t \mapsto \mu^1_{X|\mathcal{F}_{X_t}}$ belongs to a compact subset of $\mathcal{X}(\mathcal{H}_\mathcal{S}(V))$, which satisfies the assumptions introduced in Section 2.

## 3.3 Second order kernel mean embedding and maximum mean discrepancy

The $2^{nd}$ *order KME* is the point in $\mathcal{H}_\mathcal{S}(\mathcal{H}_\mathcal{S}(V))$ defined as the KME of the $1^{\text{st}}$ order predictive KME

$$\mu^2_X = \int_{x \in \mathcal{X}(\mathcal{H}_\mathcal{S}(V))} k_\mathcal{S}(\cdot, x)\mathbb{P}_{\mu^1_{X|\mathcal{F}_X}}(dx) \tag{14}$$

The $2^{nd}$ *order MMD* of $X, Y$ is the norm of the difference in $\mathcal{H}_\mathcal{S}(\mathcal{H}_\mathcal{S}(V))$ of their $2^{\text{nd}}$ order KMEs,

$$\mathcal{D}^2_\mathcal{S}(X, Y) = \left\|\mu^2_X - \mu^2_Y\right\|_{\mathcal{H}_\mathcal{S}(\mathcal{H}_\mathcal{S}(V))} \tag{15}$$

The next theorem states that the $2^{\text{nd}}$ order MMD of two stochastic processes $X, Y$ is a stronger discrepancy measure than the $1^{\text{st}}$ order MMD.

**Theorem 2.** *Given two stochastic processes $X, Y$*

$$\mathcal{D}^2_\mathcal{S}(X, Y) = 0 \iff \mathbb{P}_{X|\mathcal{F}_X} = \mathbb{P}_{Y|\mathcal{F}_Y} \tag{16}$$

*Furthermore*

$$\mathcal{D}^2_\mathcal{S}(X, Y) = 0 \implies \mathcal{D}^1_\mathcal{S}(X, Y) = 0 \tag{17}$$

*but the converse is not generally true.*

*Proof.* All proofs are given in Appendix D. □

Next we make use of Thm. 2 in the context of two-sample hypothesis testing [1, 31] for stochastic processes. In Sec. 4 we will show by means of a numerical example that the $2^{\text{nd}}$ order MMD is able to capture filtration-related information otherwise ignored by the $1^{\text{st}}$ order MMD.

---

[9]Because $\mathbb{P}_X = \mathbb{P}_{X|\mathcal{F}_{X_T}}$, all the information about the law of $X$ is contained in just the terminal point of the trajectory traced by $\mu^1_{X|\mathcal{F}_X}$ (Fig. 1).

### 3.4 A filtration-sensitive kernel two-sample test

Suppose we are given $m$ sample paths $\{x^i\}_{i=1}^m \sim X$ and $n$ sample paths $\{y^i\}_{i=1}^n \sim Y$. A classical two-sample test [1] for $X, Y$ tests a null-hypothesis

$$H_0 : \mathbb{P}_X = \mathbb{P}_Y \quad \text{against the alternative} \quad H_A : \mathbb{P}_X \neq \mathbb{P}_Y \tag{18}$$

The probability of falsely rejecting the null is called the *type I error* (and similarly the probability of falsely accepting the null is called the *type II error*). If the type I error can be bounded from above by a constant $\alpha$, then we say that the test is of level $\alpha$. In [31, Sec. 8] it is shown that rejecting the null if $\widehat{\mathcal{D}}_{\mathcal{S}}^1(X, Y)^2 > c_\alpha$ (for some $c_\alpha$ that depends on $m, n$ and $\alpha$) gives a test of level $\alpha$, where $\widehat{\mathcal{D}}_{\mathcal{S}}^1(X, Y)$ denotes the classical unbiased estimator of the 1st order MMD [1]. This choice of threshold is conservative and can be improved by using data-dependent bounds such as in permutation tests (we refer to the MMD testing literature for extra details [1, 32, 33]).

However, as discussed in Sec. 3.3 comparing the laws $\mathbb{P}_X, \mathbb{P}_Y$ via the estimator above might be insufficient to capture filtration-related information about $X, Y$. To overcome this limitation we propose instead to test the null-hypothesis

$$H_0 : \mathbb{P}_{X|\mathcal{F}_X} = \mathbb{P}_{Y|\mathcal{F}_Y} \quad \text{against the alternative} \quad H_A : \mathbb{P}_{X|\mathcal{F}_X} \neq \mathbb{P}_{Y|\mathcal{F}_Y} \tag{19}$$

Using Thm. 2, one can immediately construct a filtration-sensitive kernel two-sample test for (19) provided one can build an empirical estimator of the 2nd order MMD $\mathcal{D}_{\mathcal{S}}^2(X, Y)$. In the rest of this section we explain how to obtain such an estimator and ultimately show its consistency.

Assuming availability of $m$ sample paths $\{\widetilde{x}^i\}_{i=1}^m$ from the stochastic process $\mu_{X|\mathcal{F}_X}^1$ and $n$ sample paths $\{\widetilde{y}^i\}_{i=1}^n$ from $\mu_{Y|\mathcal{F}_Y}^1$, an estimator of the squared 2nd order MMD is given by

$$\widehat{\mathcal{D}}_{\mathcal{S}}^2(X, Y)^2 = \frac{1}{m(m-1)} \sum_{\substack{i,j=1 \\ i \neq j}}^m k_{\mathcal{S}}(\widetilde{x}^i, \widetilde{x}^j) - \frac{2}{mn} \sum_{i,j=1}^{m,n} k_{\mathcal{S}}(\widetilde{x}^i, \widetilde{y}^j) + \frac{1}{n(n-1)} \sum_{\substack{i,j=1 \\ i \neq j}}^n k_{\mathcal{S}}(\widetilde{y}^i, \widetilde{y}^j)$$

Computing this estimator boils down to evaluating the signature kernel $k_{\mathcal{S}}(\widetilde{x}, \widetilde{y})$ on sample paths $\widetilde{x} \sim \mu_{X|\mathcal{F}_X}^1$ and $\widetilde{y} \sim \mu_{Y|\mathcal{F}_Y}^1$. By Thm. 1, the signature kernel solves the following PDE

$$\frac{\partial^2 u_{\widetilde{x},\widetilde{y}}}{\partial s \partial t} = \left( \langle \widetilde{x}_{s-\delta}, \widetilde{y}_{t-\delta} \rangle_{\mathcal{H}_{\mathcal{S}}(V)} - \langle \widetilde{x}_{s-\delta}, \widetilde{y}_t \rangle_{\mathcal{H}_{\mathcal{S}}(V)} - \langle \widetilde{x}_s, \widetilde{y}_{t-\delta} \rangle_{\mathcal{H}_{\mathcal{S}}(V)} + \langle \widetilde{x}_s, \widetilde{y}_t \rangle_{\mathcal{H}_{\mathcal{S}}(V)} \right) u_{\widetilde{x},\widetilde{y}}$$

where the two derivatives in eq. (5) have been approximated by finite difference with time increment $\delta$. It remains to explain how to estimate, for any $s, t \in [0, T]$, the inner product $\langle \widetilde{x}_s, \widetilde{y}_t \rangle_{\mathcal{H}_{\mathcal{S}}(V)}$ from sample paths of $X$ and $Y$. This can be achieved using the formalism of *cross-covariance operators* [34] as thoroughly explained in Appendix A, which yields to the following approximation

$$\langle \widetilde{x}_s, \widetilde{y}_t \rangle_{\mathcal{H}_{\mathcal{S}}(V)} \approx \mathbf{k}_s^{x\top} (\mathbf{K}_{s,s}^{x,x} + m\lambda I_m)^{-1} \mathbf{K}_{T,T}^{x,y} (\mathbf{K}_{t,t}^{y,y} + n\lambda I_n)^{-1} \mathbf{k}_t^y \tag{20}$$

where $\mathbf{k}_s^x \in \mathbb{R}^m, \mathbf{k}_t^y \in \mathbb{R}^n$ are the vectors[10]

$$[\mathbf{k}_s^x]_i = k_{\mathcal{S}}(x_{[0,s]}^i, x_{[0,s]}), \quad [\mathbf{k}_t^y]_i = k_{\mathcal{S}}(y_{[0,t]}^i, y_{[0,t]})$$

and $\mathbf{K}_{s,s}^{x,x} \in \mathbb{R}^{m \times m}, \mathbf{K}_{T,T}^{x,y} \in \mathbb{R}^{n \times n}, \mathbf{K}_{t,t}^{y,y} \in \mathbb{R}^{n \times n}$ are the matrices

$$[\mathbf{K}_{s,s}^{x,x}]_{i,j} = k_{\mathcal{S}}(x_{[0,s]}^i, x_{[0,s]}^j), \quad [\mathbf{K}_{T,T}^{x,y}]_{i,j} = k_{\mathcal{S}}(x_{[0,T]}^i, y_{[0,T]}^j), \quad [\mathbf{K}_{t,t}^{y,y}]_{i,j} = k_{\mathcal{S}}(y_{[0,t]}^i, y_{[0,t]}^j)$$

and where $I_m$ (resp. $I_n$) is the $m \times m$ (resp. $n \times n$) identity matrix. The corresponding algorithm and its complexity analysis are provided in Appendix B.

The next theorem ensures that the estimator $\widehat{\mathcal{D}}_{\mathcal{S}}^2(X, Y)$ is consistent for the 2nd order MMD.

**Theorem 3.** $\widehat{\mathcal{D}}_{\mathcal{S}}^2(X, Y)$ *is a consistent estimator for the 2nd order MMD, i.e.*

$$|\widehat{\mathcal{D}}_{\mathcal{S}}^2(X, Y) - \mathcal{D}_{\mathcal{S}}^2(X, Y)| \xrightarrow{p} 0 \quad \text{as } m, n \to \infty \tag{21}$$

*with* $\{x^i\}_{i=1}^m \sim X, \{y^i\}_{i=1}^n \sim Y$ *and where convergence is in probability.*

We now iterate the procedure presented so far to define higher order KMEs and MMDs.

---

[10]Here we use the notation $x_{[0,s]}$ to denote the restriction of the path $x$ to the sub-interval $[0, s] \subset [0, T]$.

## 3.5 Higher order kernel mean embeddings and maximum mean discrepancies

One can iterate the procedure described in Sec. 3.3 and recursively define, for any $n \in \mathbb{N}_{>1}$, the $n^{th}$ *order KME of $X$* as the following point in $\mathcal{H}^n_{\mathcal{S}}(V)$

$$\mu^n_X = \int_{x \in \mathcal{X}(\mathcal{H}^{n-1}_{\mathcal{S}}(V))} k_{\mathcal{S}}(\cdot, x) \mathbb{P}_{\mu^{n-1}_{X|\mathcal{F}_X}}(dx) \tag{22}$$

where $\mu^{n-1}_{X|\mathcal{F}_X}$ is the $(n-1)^{\text{st}}$ predictive KME of $X$ and

$$\mathcal{H}^n_{\mathcal{S}}(V) = \underbrace{\mathcal{H}_{\mathcal{S}}(\mathcal{H}_{\mathcal{S}}(\ldots \mathcal{H}_{\mathcal{S}}(V)\ldots))}_{n \text{ times}} \tag{23}$$

The associated $n^{th}$ *order MMD* between two processes $X, Y$ is then defined as the norm of the difference in $\mathcal{H}^n_{\mathcal{S}}(V)$ of the two $n^{\text{th}}$ order KMEs

$$\mathcal{D}^n_{\mathcal{S}}(X, Y) = \|\mu^n_X - \mu^n_Y\|_{\mathcal{H}^n_{\mathcal{S}}(V)} \tag{24}$$

The following result generalizes Thm. 2 in that it shows that the $n^{\text{th}}$ order MMD is a stronger (i.e. finer) discrepancy measure than all the $k^{\text{th}}$ order MMDs of lower order $1 < k < n$.

**Theorem 4.** *Given two stochastic processes $X, Y$*

$$\mathcal{D}^n_{\mathcal{S}}(X, Y) = 0 \implies \mathcal{D}^k_{\mathcal{S}}(X, Y) = 0 \quad \text{for any } 1 < k < n \tag{25}$$

*but the converse is not generally true.*

Other than hypothesis testing, another important application relying on the ability of distinguishing random variables is *distribution regression* (DR) [5]. In the next section we make use of the $n^{\text{th}}$ order MMD in the setting of DR on path-valued random variables presented in [6] and propose a family of kernels on stochastic processes whose RKHSs contains richer classes of functions than the RKHS associated to the universal kernel proposed in [6].

We note that since $V$ is a Polish space (i.e., a separable, complete metric space) and the signature maps is continuous, in view of [35, Lemma 4.33] one can easily check that all RKHSs appearing in the present paper are separable Hilbert spaces by an induction argument and therefore all regular conditional distributions are well–defined.

## 3.6 Higher order distribution regression

DR on stochastic processes describes the supervised learning problem where the input is a collection of sample paths and the output is a vector of scalars [6]. Denote by $\mathcal{P}(\mathcal{X}(V))$ the set of stochastic processes with sample paths on $\mathcal{X}(V)$. Following the setup in [6], the goal is to learn a function $F : \mathcal{P}(\mathcal{X}(V)) \to \mathbb{R}$ from a training set of input-output pairs $\{(X_i, y_i)\}$ with $X_i \in \mathcal{P}(\mathcal{X}(V))$ and $y_i \in \mathbb{R}$, by means of a classical two-step procedure [16, 17, 18].

Firstly, a stochastic process $X \in \mathcal{P}(\mathcal{X}(V))$ is embedded into its KME $\mu^1_X \in \mathcal{H}_{\mathcal{S}}(V)$ via the signature kernel $k_{\mathcal{S}}$. Secondly, another function $G : \mathcal{H}_{\mathcal{S}}(V) \to \mathbb{R}$ is learnt by solving the minimization $\arg\min_{G \in \mathcal{H}_{\text{RBF}}} \sum_i \mathcal{L}(g(\mu^1_{X_i}), y_i)$, where $\mathcal{L}$ is a loss function, and $\mathcal{H}_{\text{RBF}}$ is the RKHS associated to the classical Gaussian kernel $k_{\text{RBF}} : \mathcal{H}_{\mathcal{S}}(V) \times \mathcal{H}_{\mathcal{S}}(V) \to \mathbb{R}$. This procedure materialises into a kernel on stochastic processes whose RKHS is shown to be dense in the space of functions $F : \mathcal{P}(\mathcal{X}(V)) \to \mathbb{R}$ that are continuous with respect to the weak topology [6, Thm. 3.3].

However, a class of approximators that is universal with respect to some topology is not guaranteed to well approximate functions that are discontinous with respect to that topology (but potentially continuous with respect to a finer topology). For example, financial practitioners are often interested in calibrating financial models to market data or pricing financial instruments from observations of market dynamics. These tasks can be formulated as DR problems on stochastic processes (see experiments in Sec. 4.2), but the resulting learnable functions are discontinuous with respect to the $1^{\text{st}}$ order MMD whilst being continuous with respect to the $2^{\text{nd}}$ order MMD [27]. This motivates the need to extend the kernel-based DR technique proposed in [6] to situations where the target functions are not weakly continuous, which is what Thm. 5 addresses. A function $f : \mathbb{R} \to \mathbb{R}$ is called *globally analytic with non-negative coefficients* if admits everywhere a Taylor expansion where all the coefficients are strictly positive, i.e. for any $x \in \mathbb{R}$ we have $f(x) = \sum_{i=0}^{\infty} a_i x^i$ with $a_i > 0$.

**Theorem 5.** *Let $f : \mathbb{R} \to \mathbb{R}$ be a globally analytic function with non-negative coefficients. Define the family of kernels $K_{\mathcal{S}}^n : \mathcal{P}(\mathcal{X}(V)) \times \mathcal{P}(\mathcal{X}(\mathbb{R}^d)) \to \mathbb{R}$ as follows*

$$K_{\mathcal{S}}^n(X, Y) = f(\mathcal{D}_{\mathcal{S}}^n(X, Y)), \quad n \in \mathbb{N}_{\geq 1} \tag{26}$$

*Then the RKHS associated to $K_{\mathcal{S}}^n$ is dense in the space of functions from $\mathcal{P}(\mathcal{X}(\mathbb{R}^d))$ to $\mathbb{R}$ which are continuous with respect to the $k^{th}$ order MMD for any $1 < k \leq n$.*

In Sec. 4 we will take $f(x) = \exp(-x^2/\sigma)$ with $\sigma > 0$. This result marks the end of our analysis. Next we apply our theoretical results in the contexts of two-sample testing, DR and causal inference.

## 4 Applications

Here we demonstrate the practical advantage of using $2^{nd}$ order kernel mean embeddings, and evaluate the conditional kernel mean embedding for stochastic processes on a causal discovery task. Additional experimental details can be found in Appendix C and the code is available at https://github.com/maudl3116/higherOrderKME.

### 4.1 Hypothesis testing on filtrations

We start by considering two processes $X^n$ and $X$ with transition probabilities depicted in Fig. 2. Although the laws $\mathbb{P}_n$ and $\mathbb{P}$ get arbitrarily close for large $n$, their filtrations are very different. Indeed, the two processes have different information structures available before time $t = 1$. Indeed, for any $0 < t \leq 1$, the trajectory of $X^n$ is deterministic, whilst the progression of $X$ remains random until $t = 1$. Being able to distinguish two such stochastic processes is crucial in quantitative finance: if

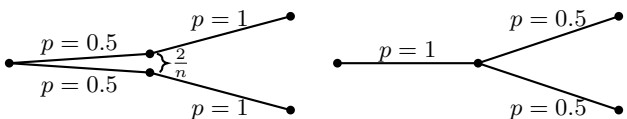

Figure 2: The supports of $\mathbb{P}_n$ (left) and $\mathbb{P}$ (right).

$\mathbb{P}_n$ and $\mathbb{P}$ are the laws of two traded assets, $\mathbb{P}_n$ gives an arbitrage opportunity. As shown in Fig. 3, the $2^{nd}$ order MMD can distinguish these two processes with similar laws ($n = 5 \cdot 10^5$) but different filtrations, while the $1^{st}$ order MMD fails to do so.

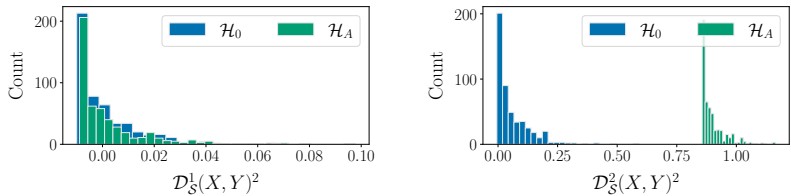

Figure 3: *Left:* Empirical distribution of the $1^{st}$ order MMD. Under $\mathcal{H}_0$ the two measures are both equal to $\mathbb{P}$ and we use 500 samples from each. Under $\mathcal{H}_A$ with $\mathbb{P}$ and $\mathbb{P}_n$ where $n = 5 \cdot 10^5$, and we use 500 samples. *Right:* Same for the $2^{nd}$ order MMD. Histograms are obtained by computing 500 independent instances of the MMD.

### 4.2 Applications of higher order distribution regression to quantitative finance

In this section we use kernel Ridge regression and support vector machine (SVM) classification equipped with the kernel $K_{\mathcal{S}}^2$ from Thm. 5 to address two real-world problems arising in quantitative finance, notably the calibration of the *rough Bergomi model* [36] and the pricing of *American options* [37]. We benchmark our filtration-sensitive kernel $K_{\mathcal{S}}^2$ against a range of kernels, including $K_{\mathcal{S}}^1$.

The rough Bergomi model is a rough volatility model [38] satisfying the following stochastic dynamics

$$dS_t = \sqrt{V_t}S_t dW_t, \quad V_t = \int_0^t K(s,t)dZ_s, \quad Z_t = \rho W_t + \sqrt{1-\rho^2}W_t' \tag{27}$$

where $W, W'$ are two independent Brownian motions and $K(s,t) = (t-s)^{h-0.5}$ where here we take $h = 0.2$. The model in eq. (27) is non-Markovian in the sense that the conditional law of $S \mid \mathcal{F}_{S_t}$ depends pathwise on the past history of the process. Of particular importance is the correct retrieval of the sign of the correlation parameter $\rho$ [39]. We consider 50 parameter values $\{\rho_i\}_{i=1}^{50}$ chosen uniformly at random from $[-1, 1]$. Each $\rho_i$ is regressed on a collection of $m = 200$ sample trajectories. We use an SVM classifier endowed with different kernels (Table 1).

One of the most studied optimal stopping problems is the pricing of an American option with a non-negative payoff function $g : \mathbb{R}^d \to \mathbb{R}$. Stock prices are assumed to follow a $d$-dimensional stochastic process $X$. The price of the corresponding option is the solution of the optimal stopping problem $\sup_\tau \mathbb{E}[g(X_\tau) \mid X_0]$, where the supremum is taken over stopping times $\tau$. Despite significant advances, pricing American options remains one of the most computationally challenging problems in financial optimization, in particular when the underlying process $X$ is non-Markovian.

Table 1: Quantitative finance examples. Average performances with standard errors in parenthesis.

| Kernel | Rough Bergomi model calibration (Acc.) | American option pricing (MSE $\times 10^{-3}$) |
|---|---|---|
| RBF | 87% (5%) | 1.07 (0.75) |
| Matérn | 87% (3%) | 2.75 (3.05) |
| $K_{\mathcal{S}}^1$ | 91% (3%) | 0.90 (0.34) |
| $K_{\mathcal{S}}^2$ | **93%** (3%) | **0.52** (0.07) |

This is the setting we consider, modelling stock prices as sample paths from fractional Brownian motion (fBm) [40] with different Hurst exponents $h \in (0, 1)$. True target prices are obtained via expensive Monte Carlo simulations [41]. We consider 25 values of $\{h_i\}_{i=1}^{25}$ sampled uniformly at random in $[0.2, 0.8]$ and use 500 samples from each fBm. As shown in Table 1 our kernel $K_{\mathcal{S}}^2$ yields the best results on both tasks (rough Bergomi model calibration and American option pricing), systematically outperforming other classical kernels as well as the kernel $K_{\mathcal{S}}^1$ introduced in [6].

### 4.3 Inferring causal graph for interacting bodies

Finally, we consider the task of recovering the causal relationships between interacting bodies solely from observations of their multidimensional trajectories. We employ the multi-body interaction simulator from [42] in order to simulate an environment where $N$ balls are connected by invisible physical relations (e.g. a spring) and describe 2D trajectories (see Fig. 4a with $N = 3$ and 2 springs). At the beginning of a simulated episode, the initial positions of the balls are generated at random, and during the episode, the balls are subject to forces with random intensity and direction. By simulating $m$ episodes we end up with $m$ sample trajectories for each of the $N$ balls. We use the kPC algorithm [13]—which relies on conditional independence testing— with the signature kernel and evaluate its ability to recover whether any two balls are connected or not. We vary $m$ and $N$ and report the results in Figs. 4b and 4c. Each experiment is run 15 times, 30% of the runs are used to chose the hyperparameters, and the reported results have been obtained on the remaining runs. We note that for finite datasets conditional independence testing is hard without additional assumptions, as discussed in [43, 44].

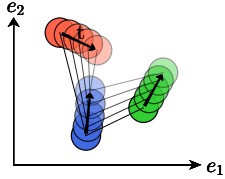

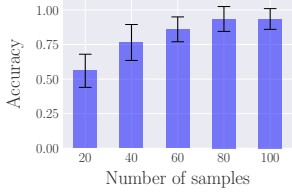

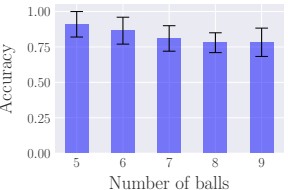

(a) 3 interacting balls describing trajectories in the 2D plane over time.

(b) Accuracy on binary classification of edges with a varying number of sample episodes (5 balls)

(c) Accuracy on binary classification of edges with a varying number of balls (100 samples)

# 5 Conclusion

In this paper, we introduced a family of higher order KMEs by conditioning a stochastic process on its filtration, generalizing the classical notion of KME. We derived an empirical estimator for the $2^{nd}$ order MMD and proved its consistency. We then proposed a filtration-sensitive kernel two-sample test and showed with simulations its ability to capture information that gets missed by the standard MMD test. In addition, we constructed a family of universal kernels on stochastic processes that allows to solve real-world calibration and optimal stopping problems in quantitative finance via Ridge regression. Finally, we designed a causal-discovery algorithm using conditional independence tests to recover the causal graph of structural dependencies among interacting bodies solely from observations of their multidimensional trajectories.

# 6 Future work

Regarding the choice of kernel hyperparameters, in the setting of two-sample tests, we can use various hyperparameter selection methods which have been proposed in the kernel literature, including approaches aiming at maximizing the test power using the signal-to-noise-ratio as an objective [45, 46, 47]. In the distribution regression setting, we made use of a cross validation approach. Developing hyperparameter tuning methodologies for higher order KMEs is an interesting future work direction and we will note that [48, 49] are certainly a good starting point for such an investigation.

Higher order KMEs have the potential to be used beyond two-sample tests and distribution regression. For example [6] recently investigated the use of the $1^{st}$ order MMD to derive an approximate Bayesian computation (ABC) algorithm for irregular time series. Another idea that is currently being investigated is using higher order MMDs as discriminators in autoregressive generative models for time series, where conditioning the future trajectories on past observations is key.

We conclude with a theoretical remark. All paths considered in the present paper are piecewise linear. Consequently, all sample paths from higher order predictive KMEs are also piecewise linear and their KMEs are well defined. Such a nice property will not hold anymore if one considered more generic continuous sample paths, because such regularity of sample paths from the corresponding higher order predictive KMEs might break as noted in [29, Remark 1]. The study of how the regularity changes by taking higher order kernel mean embeddings is an interesting direction for future work.

## Acknowledgments and Disclosure of Funding

ML and CS were respectively supported by the EPSRC grants EP/L016710/1 and EP/R513295/1. TD acknowledges support from EPSRC (EP/T004134/1), UKRI Turing AI Fellowship (EP/V02678X/1), and the Lloyd's Register Foundation programme on Data Centric Engineering through the London Air Quality project. ML, CS and TL were supported by the Alan Turing Institute under the EPSRC grant EP/N510129/1 and DataSig under the grant EP/S026347/1.

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
