# Appendix

This Appendix is organised as follows: A) using the formalism of cross-covariance operators we define an Hilbert-Schmidt conditional independence criterion for stochastic processes, and provide further details on the construction of the estimator for the $2^{\text{nd}}$ order MMD; B) we outline algorithms and their complexities for computing higher order MMDs; C) we provide further experimental details; D) we prove the theorems from the main paper.

## A  Cross-covariance operators

Covariance and cross-covariance operators on RKHSs are important concepts for modern applications of conditional KMEs [50, 10]. In this section we will use this formalism (adapted to the case of path-valued random variables) to firstly derive a criterion for conditional independence of stochastic processes and secondly provide more details on the derivation of our estimator of the $2^{\text{nd}}$ order MMD.

Let $X, Y \in \mathcal{P}(\mathcal{X}(V))$ be two stochastic processes and their joint process $(X, Y) \in \mathcal{P}(\mathcal{X}(V \oplus V))$. Define the *cross-covariance operator* $\mathcal{C}_{Y,X}$ as the following point in the tensor product of RKHSs $\mathcal{H}_{\mathcal{S}}(V) \otimes \mathcal{H}_{\mathcal{S}}(V)$

$$\mathcal{C}_{Y,X} = \mathbb{E}_{(X,Y)}[k_{\mathcal{S}}(\cdot, Y) \otimes k_{\mathcal{S}}(X, \cdot)] \tag{28}$$

or equivalently as the *Hilbert-Schmidt operator* $\mathcal{C}_{Y,X} : \mathcal{H}_{\mathcal{S}}(V) \to \mathcal{H}_{\mathcal{S}}(V)$ defined for any function $f \in \mathcal{H}_{\mathcal{S}}(V)$ as follows

$$\mathcal{C}_{Y,X}(f)(\cdot) = \int_{(x,y) \in \mathcal{X}(V \oplus V)} k_{\mathcal{S}}(\cdot, y) f(x) \mathbb{P}_{(X,Y)}(d(x,y)) \tag{29}$$

The equivalence between tensor product of RKHSs and Hilbert-Schmidt operators is given by the isomorphism $\Phi : \mathcal{H}_{\mathcal{S}}(V) \otimes \mathcal{H}_{\mathcal{S}}(V) \to \text{HS}(\mathcal{H}_{\mathcal{S}}(V), \mathcal{H}_{\mathcal{S}}(V))$ defined as follows

$$\Phi_{\mathcal{S}} \left( \sum_{\mathbf{k},\mathbf{k}'} \alpha_{\mathbf{k},\mathbf{k}'} \mathcal{S}^{(\mathbf{k})} \otimes \mathcal{S}^{(\mathbf{k}')} \right) = \sum_{\mathbf{k},\mathbf{k}'} \alpha_{\mathbf{k},\mathbf{k}'} \left\langle \cdot, \mathcal{S}^{(\mathbf{k})} \right\rangle_{\mathcal{H}_{\mathcal{S}}(V)} \mathcal{S}^{(\mathbf{k}')} \tag{30}$$

where $\mathcal{S}^{(\mathbf{k})}$ denotes the $\mathbf{k}^{\text{th}}$ element of an orthogonal basis of $\mathcal{H}_{\mathcal{S}}(V)$ and $\text{HS}(\mathcal{H}_{\mathcal{S}}(V), \mathcal{H}_{\mathcal{S}}(V))$ is the space of Hilbert-Schmidt operators from $\mathcal{H}_{\mathcal{S}}(V)$ to itself. An example of such basis is given by the *signature basis* defined for any path $x \in \mathcal{X}(V)$ and any coordinate $\mathbf{k} = (k_1, ..., k_j)$ as

$$\mathcal{S}^{\mathbf{k}}(x) = \int \ldots \int_{0 < s_1 < ... < s_j < T} dx_{s_1}^{(k_1)} \ldots dx_{s_j}^{(k_j)} \tag{31}$$

The centered version $\tilde{\mathcal{C}}_{Y,X} \in \mathcal{H}_{\mathcal{S}}(V) \otimes \mathcal{H}_{\mathcal{S}}(V)$ of the operator $\mathcal{C}_{Y,X}$ is defined as

$$\tilde{\mathcal{C}}_{Y,X} = \mathcal{C}_{Y,X} - \mu_X^1 \otimes \mu_Y^1 \tag{32}$$

Similarly let $\mathcal{C}_{X,X} \in \mathcal{H}_{\mathcal{S}}(V) \otimes \mathcal{H}_{\mathcal{S}}(V)$ be the following covariance operator

$$\mathcal{C}_{X,X} = \mathbb{E}_X[k_{\mathcal{S}}(X, \cdot) \otimes k_{\mathcal{S}}(X, \cdot)] \tag{33}$$

or equivalently $\mathcal{C}_{X,X} \in \text{HS}(\mathcal{H}_{\mathcal{S}}(V), \mathcal{H}_{\mathcal{S}}(V))$

$$\mathcal{C}_{X,X}(f)(\cdot) = \int_{x \in \mathcal{X}(V)} k_{\mathcal{S}}(\cdot, x) f(x) \mathbb{P}_X(dx) \tag{34}$$

Under the assumption that for any $f \in \mathcal{H}_{\mathcal{S}}(V)$ the function $x \mapsto \mu_{f(Y)|X=x}^1$ from $\mathcal{X}(V)$ to $\mathbb{R}$ is in $\mathcal{H}_{\mathcal{S}}(V)$, the authors in [10, 50] showed that

$$\mu_{Y|X}^1 = \mathcal{C}_{Y,X} \mathcal{C}_{X,X}^{-1} \tag{35}$$

However, this assumption might not hold in general [10, 50]. This technical issue can be circumvented by resorting to a regularized version of eq. (35): this yields to

$$\mu_{Y|X}^1 \approx \mathcal{C}_{Y,X}(\mathcal{C}_{X,X} + \lambda I_{\mathcal{H}_{\mathcal{S}}(V)})^{-1}, \quad \lambda > 0 \tag{36}$$

where $I_{\mathcal{H}_{\mathcal{S}}(V)}$ is the identity map from $\mathcal{H}_{\mathcal{S}}(V)$ to itself. Under some mild conditions the empirical estimator of eq. (36) is equal to the empirical estimator of eq. (35) [9, Thm. 8]. In particular, one has

$$\hat{\mu}^1_{Y|X=x} = \mathbf{k}^{x\top}(\mathbf{K}^{x,x} + m\lambda I_m)^{-1}\mathbf{k}^y(\cdot) \tag{37}$$

based on sample paths $\{(x^i, y^i)\}_{i=1}^m$ from the joint $(X, Y)$ where $\mathbf{k}^x$, $\mathbf{K}^{x,x}$ and $\mathbf{k}^y(\cdot)$ are such that

$$[\mathbf{k}^x]_i = k_{\mathcal{S}}(x^i, x) \quad [\mathbf{K}^{x,x}]_{i,j} = k_{\mathcal{S}}(x^i, x^j) \quad \mathbf{k}^y(\cdot) = [k_{\mathcal{S}}(y_1, \cdot), \ldots, k_{\mathcal{S}}(y_m, \cdot)]^\top$$

Cross-covariance operators have been used to define kernel-based measures of conditional dependence, as we shall discuss in the next section.

## A.1 Hilbert-Schmidt Conditional Independence Criterion for stochastic processes

Multiple measures of conditional dependence have been proposed in the literature [14, 13, 12]. In this section, we follow [12] to define a nonparametric conditional dependence measure for stochastic processes, based on the *conditional cross-covariance operator* $\tilde{\mathcal{C}}_{Y,X|Z} : \mathcal{H}_{\mathcal{S}}(V) \to \mathcal{H}_{\mathcal{S}}(V)$,

$$\tilde{\mathcal{C}}_{Y,X|Z} = \tilde{\mathcal{C}}_{Y,X} - \tilde{\mathcal{C}}_{Y,Z}\tilde{\mathcal{C}}_{Z,Z}^{-1}\tilde{\mathcal{C}}_{Z,X} \tag{38}$$

The squared Hilbert-Schmidt norm $H_{YX|Z} := \|\tilde{\mathcal{C}}_{(Y,Z),X|Z}\|^2_{\text{HS}}$ can be used as measure of conditional dependence of stochastic processes. Since the signature kernel $k_{\mathcal{S}}$ is characteristic, it follows that $X \perp\!\!\!\perp Y \mid Z \iff H_{YX|Z} = 0$ [12].

Given $m$ sample paths $\{(x_i, y_i, z_i)\}_{i=1}^m$ from the joint distribution of $(X, Y, Z)$, let $K^x$, $K^y$ and $K^z$ be the Gram matrices with entries,

$$[K^x]_{i,j} = k_{\mathcal{S}}(x_i, x_j) \quad [K^y]_{i,j} = k_{\mathcal{S}}(y_i, y_j) \quad [K^z]_{i,j} = k_{\mathcal{S}}(z_i, z_j)$$

An empirical estimator of the kernel conditional dependence measure $H_{YX|Z}$ is then given by,

$$\widehat{H}_{YX|Z} = \tfrac{1}{m^2}\left\{\text{tr}(\tilde{K}^x\tilde{K}^y) - 2\text{tr}(\tilde{K}^x\tilde{K}^z(\tilde{K}^z_\epsilon)^{-2}\tilde{K}^z\tilde{K}^y) + \text{tr}(\tilde{K}^x\tilde{K}^z(\tilde{K}^z_\epsilon)^{-2}\tilde{K}^z\tilde{K}^y\tilde{K}^z(\tilde{K}^z_\epsilon)^{-2}\tilde{K}^z)\right\}$$

where $\tilde{K}^z_\epsilon = \tilde{K}^z + \epsilon I_m$ and $\tilde{K}^x, \tilde{K}^y, \tilde{K}^z$ are the centered versions of the matrices $K^x, K^y$ and $K^z$,

$$\tilde{K}^x = HK^xH \quad \tilde{K}^y = HK^yH \quad \tilde{K}^z = HK^zH$$

with $H = I_m - m^{-1}\mathbf{1}_m$ and $\mathbf{1}_m$ the $m \times m$ matrix with all entries set to 1. This estimator can be used as a test statistic for testing whether $X$ and $Y$ are independent given $Z$. However, it is not known how to analytically compute the null distribution of the test statistic, and permutation tests are typically used. In Sec. 4.3 we use this measure of conditional dependence as part of the kPC algorithm to infer causal relationships between multidimensional stochastic processes. We provide more details in Appendix C.

## A.2 Construction of the estimator for the second order MMD $\mathcal{D}^2_{\mathcal{S}}$

As discussed in the main paper, the estimation of the 2$^{\text{nd}}$ order MMD, required the ability to compute inner products of the form $\langle \tilde{x}_s, \tilde{y}_t \rangle$ in $\mathcal{H}_{\mathcal{S}}(V)$. Here, we provide more details on the approximation that we have used in eq. (20), also restated below,

$$\langle \tilde{x}_s, \tilde{y}_t \rangle_{\mathcal{H}_{\mathcal{S}}(V)} \approx \mathbf{k}^{x\top}_s(\mathbf{K}^{x,x}_{s,s} + m\lambda I_m)^{-1}\mathbf{K}^{x,y}_{T,T}(\mathbf{K}^{y,y}_{t,t} + n\lambda I_n)^{-1}\mathbf{k}^y_t$$

where $\tilde{x}$ and $\tilde{y}$ are sample paths from the processes $\mu^1_{X|\mathcal{F}_X}$ and $\mu^1_{Y|\mathcal{F}_Y}$ respectively. In particular,

$$\tilde{x}_s = \mu^1_{X|x_{[0,s]}} \quad \text{and} \quad \tilde{y}_t = \mu^1_{Y|y_{[0,t]}}$$

As discussed at the beginning of this section, their empirical estimators are constructed from $m$ samples $\{x^i\}_{i=1}^m$ from $X$ and $n$ samples $\{y^j\}_{j=1}^n$ from $Y$ respectively

$$\tilde{x}_s \approx \mathbf{k}^{x\top}_s(\mathbf{K}^{x,x}_{s,s} + m\lambda I_m)^{-1}\mathbf{k}^x(\cdot) \quad \text{and} \quad \tilde{y}_t \approx \mathbf{k}^{y\top}_t(\mathbf{K}^{y,y}_{t,t} + n\lambda I_n)^{-1}\mathbf{k}^y(\cdot)$$

where $\mathbf{k}_s^x$, $\mathbf{K}_{s,s}^{x,x}$, $\mathbf{k}^x(\cdot)$ and $\mathbf{k}_t^y$, $\mathbf{K}_{t,t}^{y,y}$ and $\mathbf{k}^y(\cdot)$ are defined by,

$$[\mathbf{k}_s^x]_i = k_{\mathcal{S}}(x_{[0,s]}^i, x_{[0,s]}) \quad [\mathbf{K}_{s,s}^{x,x}]_{i,j} = k_{\mathcal{S}}(x_{[0,s]}^i, x_{[0,s]}^j) \quad \mathbf{k}^x(\cdot) = [k_{\mathcal{S}}(x_1, \cdot), \dots, k_{\mathcal{S}}(x_m, \cdot)]^\top$$

$$[\mathbf{k}_t^y]_i = k_{\mathcal{S}}(y_{[0,t]}^i, y_{[0,t]}) \quad [\mathbf{K}_{t,t}^{y,y}]_{i,j} = k_{\mathcal{S}}(y_{[0,t]}^i, y_{[0,t]}^j) \quad \mathbf{k}^y(\cdot) = [k_{\mathcal{S}}(y_1, \cdot), \dots, k_{\mathcal{S}}(y_n, \cdot)]^\top$$

Alternatively, we can write $\widetilde{x}_s \approx \sum_{i=1}^m \alpha_i k_{\mathcal{S}}(x^i, \cdot)$ with $\boldsymbol{\alpha} = (\mathbf{K}_{s,s}^{x,x} + m\lambda I_m)^{-1} \mathbf{k}_s^x$. Similarly we have $\widetilde{y}_t \approx \sum_{j=1}^n \beta_j k_{\mathcal{S}}(\cdot, y^j)$ with $\boldsymbol{\beta} = (\mathbf{K}_{t,t}^{y,y} + n\lambda I_n)^{-1} \mathbf{k}_t^y$. Therefore, the inner product between $\widetilde{x}_s$ and $\widetilde{y}_t$ can be approximated as follows,

$$
\begin{aligned}
\langle \widetilde{x}_s, \widetilde{y}_t \rangle_{\mathcal{H}_{\mathcal{S}}(V)} &\approx \sum_{i=1}^m \sum_{j=1}^n \alpha_i \beta_j k_{\mathcal{S}}(x^i, y^j) \\
&= \boldsymbol{\alpha}^\top \mathbf{K}_{T,T}^{x,y} \boldsymbol{\beta} \\
&= \mathbf{k}_s^{x\top} (\mathbf{K}_{s,s}^{x,x} + m\lambda I_m)^{-1} \mathbf{K}_{T,T}^{x,y} (\mathbf{K}_{t,t}^{y,y} + n\lambda I_n)^{-1} \mathbf{k}_t^y
\end{aligned}
$$

where $\mathbf{K}_{T,T}^{x,y} \in \mathbb{R}^{m \times n}$ with $[\mathbf{K}_{T,T}^{x,y}]_{i,j} = k_{\mathcal{S}}(x^i, y^j)$. Next we outline the algorithm to compute $\hat{\mathcal{D}}_{\mathcal{S}}^2$.

# B  Algorithms

In this section, we provide algorithms to compute the empirical estimator $\hat{\mathcal{D}}_{\mathcal{S}}^k$ for the $k^{\text{th}}$ order MMD, which rely on the ability to evaluate the signature kernel $k_{\mathcal{S}}(x, y)$ where $x$ and $y$ are two paths taking their values in the Hilbert space $\mathcal{H}^{k-1}(V)$. Following [22, Sec. 3.1.] we use an explicit finite difference scheme to approximate the PDE solution $u_{x,y}$ on a grid $\mathcal{P}$ of size $P \times Q$,

$$\mathcal{P} = \{0 = s_1 < s_2 < \dots < s_P = T\} \times \{0 = t_1 < t_2 < \dots < t_Q = T\}$$

Writing $u_{x,y}(s_i, t_j) = u_{i,j}$ to make the notation more concise, we use an update rule of the form,

$$u_{i+1,j+1} = f(u_{i,j+1}, u_{i+1,j}, u_{i,j}, M_{i,j}), \quad M_{i,j} = \langle x_{s_{i+1}} - x_{s_i}, y_{t_{j+1}} - y_{t_j} \rangle_{\mathcal{H}_{\mathcal{S}}^{k-1}(V)}$$

Hence, computing $k_{\mathcal{S}}(x, y)$ consists in forming the $(P-1) \times (Q-1)$ matrix $M$ such that,

$$[M]_{i,j} = \langle x_{s_{i+1}} - x_{s_i}, y_{t_{j+1}} - y_{t_j} \rangle_{\mathcal{H}_{\mathcal{S}}^{k-1}(V)}$$

and iteratively applying the update rule as outlined in Alg. 1. Besides, in Alg. 1 we distinguish the case where the solution $u$ on the entire grid is returned, and the case where only the solution at the final times $(s_P, t_Q) = (T, T)$ is returned, which corresponds to the value of the kernel $k_{\mathcal{S}}(x, y)$. The runtime complexity to solve one PDE is $\mathcal{O}(PQ)$. We make use of parallelization strategy to drastically speed-up the PDE solver on CUDA-enabled GPUs.

## B.1  Algorithm for the $1^{\text{st}}$ order MMD

In this section we provide the algorithm to compute an empirical estimator of the $1^{\text{st}}$ order MMD. This way, we introduce subroutines (Alg. 1 and Alg. 2) for the estimator of the $2^{\text{nd}}$ order MMD. We assume that $m = n$ and $P = Q$ to simplify the final runtime complexities of the algorithms.

Let $\{x^i\}_{i=1}^m \sim X$ and $\{y^j\}_{j=1}^n \sim Y$. An unbiased estimator of the $1^{\text{st}}$ order MMD [1] reads as,

$$\hat{\mathcal{D}}_{\mathcal{S}}^1(X, Y) = \frac{1}{m(m-1)} \sum_{\substack{i,j=1 \\ i \neq j}}^m k_{\mathcal{S}}(x^i, x^j) - \frac{2}{mn} \sum_{i,j=1}^{m,n} k_{\mathcal{S}}(x^i, y^j) + \frac{1}{n(n-1)} \sum_{\substack{i,j=1 \\ i \neq j}}^n k_{\mathcal{S}}(y^i, y^j)$$

Hence, in order to compute this estimator, we need to form the following three Gram matrices $\mathbf{G}_{X,X}^1 \in \mathbb{R}^{m \times m}$, $\mathbf{G}_{X,Y}^1 \in \mathbb{R}^{m \times n}$ and $\mathbf{G}_{Y,Y}^1 \in \mathbb{R}^{n \times n}$ such that,

$$[\mathbf{G}_{X,X}^1]_{i,j} = k_{\mathcal{S}}(x^i, x^j) \quad [\mathbf{G}_{X,Y}^1]_{i,j} = k_{\mathcal{S}}(x^i, y^j) \quad [\mathbf{G}_{Y,Y}^1]_{i,j} = k_{\mathcal{S}}(y^i, y^j)$$

As explained at the begining of this section (and outlined in Alg. 2), this consists in two steps. Taking $\mathbf{G}_{X,Y}^1$ for example, first one forms $m \times n$ matrices of size $(P-1) \times (Q-1)$ each of the form,

$$[M]_{p,q} = \langle x_{s_{p+1}}^i - x_{s_p}^i, y_{t_{q+1}}^j - y_{t_q}^j \rangle_V$$

and then one solves $m \times n$ PDEs with Alg. 1. The full procedure is summarized in Alg. 3, which has time complexity $\mathcal{O}(dm^2 P^2)$ where $d$ is the number of coordinates of the paths $x$ and $y$.

---

**Algorithm 1** PDESolve $\qquad\qquad\qquad\qquad\qquad\qquad\qquad\qquad\qquad\qquad\qquad\qquad\qquad\qquad\mathcal{O}(P^2)$

---

1: **Input:** matrix $M \in \mathbb{R}^{P \times Q}$, full $\in \{\text{True}, \text{False}\}$
2: **Output:** full solution $u \in \mathbb{R}^{P \times Q}$ with $u[p,q] = k_{\mathcal{S}}(x_{[0,s_p]}, y_{[0,t_q]})$ or $u[-1,-1] = k_{\mathcal{S}}(x,y)$

3: $u[1,:] \leftarrow 1$
4: $u[:,1] \leftarrow 1$
5: **for** $p$ from 1 to $P-1$ **do**
6: $\quad$ **for** $q$ from 1 to $Q-1$ **do**
7: $\qquad u[p+1, q+1] \leftarrow f(u[p, q+1], u[p+1, q], u[p, q], M[p, q])$
8: **if** full **then return** $u$ **else return** $u[-1,-1]$

---

---

**Algorithm 2** FirstOrderGram $\qquad\qquad\qquad\qquad\qquad\qquad\qquad\qquad\qquad\qquad\qquad\qquad\mathcal{O}(dm^2P^2)$

---

1: **Input:** sample paths $\{x^i\}_{i=1}^m \sim X$ and $\{y^j\}_{j=1}^n \sim Y$, full $\in \{\text{True}, \text{False}\}$
2: **Output:** $G \in \mathbb{R}^{m \times n \times P \times Q}$ where $G[i,j,p,q] = k_{\mathcal{S}}(x^i_{[0,s_p]}, y^j_{[0,t_q]})$ or $G[:,:,-1,-1]$

3: $M[i,j,p,q] \leftarrow \langle x^i_{s_p}, y^j_{t_q} \rangle \quad \forall i \in \{1,\ldots,m\}, j \in \{1,\ldots,n\}, p \in \{1,\ldots,P\}, q \in \{1,\ldots,Q\}$
4: $M \leftarrow M[:,:,1:,1:] + M[:,:,:-1,:-1] - M[:,:,1:,:-1] - M[:,:,:-1,1:]$
5: $G[i,j] \leftarrow \text{PDESolve}(M[i,j]), \quad \forall i \in \{1,\ldots,m\}, j \in \{1,\ldots,n\}$
6: **if** full **then return** $G$ **else return** $G[:,:,-1,-1]$

---

### B.2 Algorithm for the $2^{\text{nd}}$ order MMD

In the main paper, we derived the following estimator of the $2^{\text{nd}}$ order MMD,

$$\widehat{\mathcal{D}}_{\mathcal{S}}^2(X,Y) = \frac{1}{m(m-1)} \sum_{\substack{i,j=1 \\ i \neq j}}^{m} k_{\mathcal{S}}(\widetilde{x}^i, \widetilde{x}^j) - \frac{2}{mn} \sum_{i,j=1}^{m,n} k_{\mathcal{S}}(\widetilde{x}^i, \widetilde{y}^j) + \frac{1}{n(n-1)} \sum_{\substack{i,j=1 \\ i \neq j}}^{n} k_{\mathcal{S}}(\widetilde{y}^i, \widetilde{y}^j)$$

Compared to the $1^{\text{st}}$ order MMD, in order to compute this estimator, as outlined in Alg. 8, we need to form the following three Gram matrices $\mathbf{G}_{X,X}^2 \in \mathbb{R}^{m \times m}$, $\mathbf{G}_{X,Y}^2 \in \mathbb{R}^{m \times n}$ and $\mathbf{G}_{Y,Y}^2 \in \mathbb{R}^{n \times n}$,

$$[\mathbf{G}_{X,X}^2]_{i,j} = k_{\mathcal{S}}(\widetilde{x}^i, \widetilde{x}^j) \quad [\mathbf{G}_{X,Y}^2]_{i,j} = k_{\mathcal{S}}(\widetilde{x}^i, \widetilde{y}^j) \quad [\mathbf{G}_{Y,Y}^2]_{i,j} = k_{\mathcal{S}}(\widetilde{y}^i, \widetilde{y}^j)$$

As outlined in Alg. 7, this consists in two steps. Taking $\mathbf{G}_{X,Y}^2$ for example, first one forms $m \times n$ matrices of size $(P-1) \times (Q-1)$ each of the form,

$$[M]_{p,q} = \langle \widetilde{x}_{s_{p+1}} - \widetilde{x}_{s_p}, \widetilde{y}_{t_{q+1}} - \widetilde{y}_{t_q} \rangle_{\mathcal{H}_{\mathcal{S}}(V)}$$

(see Alg. 6) and then one solves $m \times n$ PDEs with Alg. 1. This is summarized in Alg. 8, which has time complexity $\mathcal{O}((d+m)m^2P^2)$ where $d$ is the number of coordinates of the paths $x$ and $y$.

---

**Algorithm 3** FirstOrderMMD $\qquad\qquad\qquad\qquad\qquad\qquad\qquad\qquad\qquad\qquad\qquad\qquad\mathcal{O}(dm^2P^2)$

---

1: **Input:** sample paths $\{x^i\}_{i=1}^m \sim X$ and $\{y^j\}_{j=1}^n \sim Y$
2: **Output:** an empirical estimator of the $1^{\text{st}}$ order MMD between $X$ and $Y$

3: $G_{XX}^1 \leftarrow \text{FirstOrderGram}(\{x^i\}_{i=1}^m, \{x^i\}_{i=1}^m, \text{full} = \text{False})$
4: $G_{XY}^1 \leftarrow \text{FirstOrderGram}(\{x^i\}_{i=1}^m, \{y^j\}_{j=1}^n, \text{full} = \text{False})$
5: $G_{YY}^1 \leftarrow \text{FirstOrderGram}(\{y^j\}_{j=1}^n, \{y^j\}_{j=1}^n, , \text{full} = \text{False})$

6: **return** $\text{avg}(G_{XX}^1) - 2 * \text{avg}(G_{XY}^1) + \text{avg}(G_{YY}^1)$

---

---

**Algorithm 4** SecondOrderGram $\hspace{6cm}$ $\mathcal{O}(P^2 m^3)$

---

1: **Input:** $G_{XX}, G_{XY}, G_{YY}$ with $G_{XY}[i,j,p,q] = k_{\mathcal{S}}(x^i_{[0,s_p]}, y^j_{[0,t_q]})$ and hyperparameter $\lambda$.
2: **Output:** an empirical estimator of $G^2_{X,Y} \in \mathbb{R}^{m \times n}$, where $G^2_{X,Y}[i,j] = k_{\mathcal{S}}(\widetilde{x}^i, \widetilde{y}^j)$

3: $M \leftarrow \mathsf{InnerProdPredCondKME}(G_{XX}, G_{XY}, G_{YY}, \lambda)$
4: $M \leftarrow M[:,:,1:,1:] + M[:,:,:-1,:-1] - M[:,:,1:,:-1] - M[:,:,:-1,1:]$
5: $G^2_{XY}[i,j] \leftarrow \mathsf{PDESolve}(M[i,j]), \quad \forall i \in \{1,\dots,m\}, \forall j \in \{1,\dots,n\}$
6: **return** $G^2_{XY}$

---

---

**Algorithm 5** SecondOrderMMD $\hspace{4cm}$ $\mathcal{O}(dm^2 P^2 + P^2 m^3)$

---

1: **Input:** sample paths $\{x^i\}^m_{i=1} \sim X$ and $\{y^j\}^n_{j=1} \sim Y$, hyperparameter $\lambda$.
2: **Output:** an empirical estimator of the 2$^{\text{nd}}$ order MMD between $X$ and $Y$

3: $G^1_{XX} \leftarrow \mathsf{FirstOrderGram}(\{x^i\}^m_{i=1}, \{x^i\}^m_{i=1}, \mathsf{full} = \mathsf{True})$
4: $G^1_{XY} \leftarrow \mathsf{FirstOrderGram}(\{x^i\}^m_{i=1}, \{y^j\}^n_{j=1}, \mathsf{full} = \mathsf{True})$
5: $G^1_{YY} \leftarrow \mathsf{FirstOrderGram}(\{y^j\}^n_{j=1}, \{y^j\}^n_{j=1}, \mathsf{full} = \mathsf{True})$

6: $G^2_{XX} \leftarrow \mathsf{SecondOrderGram}(G^1_{XX}, G^1_{XX}, G^1_{XX}, \lambda)$
7: $G^2_{XY} \leftarrow \mathsf{SecondOrderGram}(G^1_{XX}, G^1_{XY}, G^1_{YY}, \lambda)$
8: $G^2_{YY} \leftarrow \mathsf{SecondOrderGram}(G^1_{YY}, G^1_{YY}, G^1_{YY}, \lambda)$

9: **return** $\mathrm{avg}(G^2_{XX}) - 2 * \mathrm{avg}(G^2_{XY}) + \mathrm{avg}(G^2_{YY})$

---

---

**Algorithm 6** InnerProdPredCondKME $\hspace{5cm}$ $\mathcal{O}(P^2 m^3)$

---

1: **Input:** three Gram matrices $G_{XX}, G_{XY}, G_{YY}$ and hyperparameter $\lambda$
2: **Output:** returns an empirical estimator of $M \in \mathbb{R}^{m \times n \times P \times Q}$ where $M[i,j,p,q] = \langle \widetilde{x}^i_{s_p}, \widetilde{y}^j_{t_q} \rangle$
3: $W_X[:,:,p] \leftarrow (G_{XX}[:,:,p,p] + m\lambda I)^{-1}, \quad \forall p \in \{1,\dots,P\}$
4: $W_Y[:,:,q] \leftarrow (G_{YY}[:,:,q,q] + n\lambda I)^{-1}, \quad \forall q \in \{1,\dots,Q\}$
5: **for** $p$ from 1 to $P$ **do**
6: $\quad$ **for** $q$ from 1 to $Q$ **do**
7: $\quad\quad M[:,:,p,q] \leftarrow G_{XX}[:,:,p,p]^T W_X[:,:,p] G_{XY}[:,:,-1,-1] W_Y[:,:,q] G_{YY}[:,:,q,q]$

---

## B.3 Algorithm for higher order MMDs

Now, we generalize the procedure in Appendix B.2 for computing an estimator of $\mathcal{D}^{k+1}_{\mathcal{S}}$ when $k > 1$,

$$\widehat{\mathcal{D}}^{k+1}_{\mathcal{S}}(X,Y) = \frac{1}{m(m-1)} \sum_{\substack{i,j=1 \\ i \neq j}}^{m} k_{\mathcal{S}}(\widetilde{x}^{k,i}, \widetilde{x}^{k,j}) - \frac{2}{mn} \sum_{i,j=1}^{m,n} k_{\mathcal{S}}(\widetilde{x}^{k,i}, \widetilde{y}^{k,j}) + \frac{1}{n(n-1)} \sum_{\substack{i,j=1 \\ i \neq j}}^{n} k_{\mathcal{S}}(\widetilde{y}^{k,i}, \widetilde{y}^{k,j}),$$

where $\widetilde{x}^{k,i}$ and $\widetilde{y}^{k,j}$ denote sample paths from the processes $\mu^k_{X|\mathcal{F}_X}$ and $\mu^k_{Y|\mathcal{F}_Y}$ respectively. In order to compute this estimator, as outlined in Alg. 8, we need to form the following three Gram matrices $\mathbf{G}^{k+1}_{X,X} \in \mathbb{R}^{m \times m}$, $\mathbf{G}^{k+1}_{X,Y} \in \mathbb{R}^{m \times n}$ and $\mathbf{G}^{k+1}_{Y,Y} \in \mathbb{R}^{n \times n}$,

$$[\mathbf{G}^{k+1}_{X,X}]_{i,j} = k_{\mathcal{S}}(\widetilde{x}^{k,i}, \widetilde{x}^{k,j}) \quad [\mathbf{G}^{k+1}_{X,Y}]_{i,j} = k_{\mathcal{S}}(\widetilde{x}^{k,i}, \widetilde{y}^{k,j}) \quad [\mathbf{G}^{k+1}_{Y,Y}]_{i,j} = k_{\mathcal{S}}(\widetilde{y}^{k,i}, \widetilde{y}^{k,j})$$

As outlined in Alg. 7, this consists in two steps. Taking $\mathbf{G}^{k+1}_{X,Y}$ for example, first one forms $m \times n$ matrices of size $(P-1) \times (Q-1)$ each of the form,

$$[M]_{p,q} = \langle \widetilde{x}^k_{s_{p+1}} - \widetilde{x}^k_{s_p}, \widetilde{y}^k_{t_{q+1}} - \widetilde{y}^k_{t_q} \rangle_{\mathcal{H}^k_{\mathcal{S}}(V)}$$

(see Alg. 6) and then one solves $m \times n$ PDEs with Alg. 1. This is summarized in Alg. 8, which has time complexity $\mathcal{O}((d + km)m^2 P^2)$ where $d$ is the number of coordinates of the paths $x$ and $y$.

**Algorithm 7** HigherOrderGram $\qquad\qquad\qquad\qquad\qquad\qquad\qquad\qquad\qquad\qquad\qquad \mathcal{O}(P^2 m^3)$

1: **Input:** $G^k_{XX}, G^k_{XY}, G^k_{YY}$ with $G^k_{XY}[i,j,p,q] = k_\mathcal{S}(\widetilde{x}^{k-1,i}_{[0,s_p]}, \widetilde{y}^{k-1,j}_{[0,t_q]})$ and hyperparameter $\lambda$.
2: **Output:** an estimator of $G^{k+1}_{X,Y} \in \mathbb{R}^{m \times n \times P \times Q}$, where $G^{k+1}_{X,Y}[i,j,p,q] = k_\mathcal{S}(\widetilde{x}^{k,i}_{[0,s_p]}, \widetilde{y}^{k,j}_{[0,t_q]})$

3: $M \leftarrow \mathsf{InnerProdPredCondKME}(G^k_{XX}, G^k_{XY}, G^k_{YY}, \lambda)$
4: $M \leftarrow M[:,:,1:,1:] + M[:,:,:-1,:-1] - M[:,:,1:,:-1] - M[:,:,:-1,1:]$
5: $G^{k+1}_{XY}[i,j] \leftarrow \mathsf{PDESolve}(M[i,j], \mathsf{full} = \mathrm{True}), \quad \forall i \in \{1, \dots, m\}, \forall j \in \{1, \dots, n\}$
6: **return** $G^{k+1}_{XY}$

---

**Algorithm 8** HigherOrderMMD $\qquad\qquad\qquad\qquad\qquad\qquad \mathcal{O}(dm^2 P^2 + (k-1)P^2 m^3)$

1: **Input:** sample paths $\{x^i\}^m_{i=1} \sim X$ and $\{y^j\}^n_{j=1} \sim Y$, hyperparameter $\lambda$, order $k$.
2: **Output:** an empirical estimator of the $k^{\mathrm{th}}$ order MMD between $X$ and $Y$

3: $G_{XX} \leftarrow \mathsf{FirstOrderGram}(\{x^i\}^m_{i=1}, \{x^i\}^m_{i=1}, \mathsf{full} = \mathrm{True})$
4: $G_{XY} \leftarrow \mathsf{FirstOrderGram}(\{x^i\}^m_{i=1}, \{y^j\}^n_{j=1}, \mathsf{full} = \mathrm{True})$
5: $G_{YY} \leftarrow \mathsf{FirstOrderGram}(\{y^j\}^n_{j=1}, \{y^j\}^n_{j=1}, \mathsf{full} = \mathrm{True})$

6: **for** order from 2 to $k$ **do**
7: $\quad G^{\mathsf{new}}_{XX} \leftarrow \mathsf{HigherOrderGram}(G_{XX}, G_{XX}, G_{XX}, \lambda)$
8: $\quad G^{\mathsf{new}}_{XY} \leftarrow \mathsf{HigherOrderGram}(G_{XX}, G_{XY}, G_{YY}, \lambda)$
9: $\quad G^{\mathsf{new}}_{YY} \leftarrow \mathsf{HigherOrderGram}(G_{YY}, G_{YY}, G_{YY}, \lambda)$
10: $\quad G_{XX}, G_{XY}, G_{YY} \leftarrow G^{\mathsf{new}}_{XX}, G^{\mathsf{new}}_{XY}, G^{\mathsf{new}}_{YY}$

11: **return** $\mathrm{avg}(G_{XX}[:,:,-1,-1]) - 2 * \mathrm{avg}(G_{XY}[:,:,-1,-1]) + \mathrm{avg}(G_{YY}[:,:,-1,-1])$

---

# C   Experimental details

We start with further experimental details for the applications of higher order distribution regression to quantitative finance (Sec. 4.2), where we consider the problem of optimally stopping fractional Brownian motions with different hurst exponents.

## C.1   Rough volatility

Rough volatility models constitute a class of models that are empirically well-tailored to fit observed implied market volatilities in the context of option pricing for short maturity assets. The basic model for option pricingis called the Black-Scholes model in which the volatility is assumed to be constant. Stochastic volatility models are extensions of the Black-Scholes model to the case where the volatility is itself stochastic. The main shortcoming of such stochastic volatility models is that they are able to capture the true steepness of the implied volatility smile close to maturity (see [36] for extra details). This is where rough volatility models become useful. Among them, the rough Bergomi model introduced by [36], stood out for its ability to explain implied volatility and other phenomena related to European options.

## C.2   Higher order distribution regression

**Data**   We use the data generator from `https://github.com/HeKrRuTe/OptStopRandNN` to simulate sample paths from $X$ a fractional Brownian motion (fBm) and obtain the solution of the optimal stopping time problem $\sup_\tau \mathbb{E}[g(X_\tau)|X_0]$. We note that although fBm is not typically used as a stock price model in quantitative finance, it is nevertheless considered a respected challenging example for optimal stopping algorithms [51, 52].

**Models**   We use a kernel Ridge regressor with different distribution regression kernels. Each is of the form $K(X,Y) = \exp(-\mathcal{D}(X,Y)^2/\sigma^2)$ where $\mathcal{D}(X,Y)$ is a maximum mean discrepancy. The models $K^1_\mathcal{S}$ and $K^2_\mathcal{S}$ correspond to the 1st and 2nd order maximum mean discrepancies $\mathcal{D}^1_\mathcal{S}$ and $\mathcal{D}^2_\mathcal{S}$. We consider two other baselines (Matérn and RBF) for which the MMD is computed using the

Matern 3/2 covariance function $k_{\text{mat32}}$, and the RBF covariance function $k_{\text{rbf}}$,

$$k_{\text{mat32}}(x,y) = \left(1 + \frac{\sqrt{3}}{\gamma^2}\|x-y\|\right)\exp\left(-\frac{\sqrt{3}}{\gamma^2}\|x-y\|\right), \quad k_{\text{rbf}}(x,y) = \exp\left(-\frac{\|x-y\|^2}{\gamma^2}\right)$$

All models are run 3 times. The hyperparameters of all models are selected by cross-validation via a grid search on the training set (70% of the data selected at random) of each run.

### C.2.1 Inferring causal graph for interacting bodies

We provide further details for the last application (Sec. 4.3) where the task is to infer whether any two bodies are connected by a spring from multiple observations of their 2D trajectories.

**Data** We adapt the Pymunk simulator from [42] publicly available at `https://github.com/pairlab/v-cdn`. For each pair of balls, there is a one-half probability that they are connected by nothing, or a spring. For each graph we run multiple episodes each of 20 time steps. At the beginning of each episode, we randomly assign the balls in different positions. The stiffness of the spring relation is set to 20, and we randomly sample the rest length between $[20, 120]$.

**Causal discovery algorithm** The PC algorithm [53] uses conditional independence tests to generate a causal graph from a dataset. The PC algorithm consists in two stages. The first stage, referred to as the *skeleton phase*, consists in finding the structure of the causal graph. In the second stage, the edges are oriented by repetitively applying orientation rules. In the multi-body interaction example, we only need to perform the skeleton phase, which is sketched hereafter,

1. Start with a complete graph
2. For each $X$ and $Y$ which are still connected. If there is a third variable $Z_1$ connected to $X$ or $Y$, such that $X \perp\!\!\!\perp Y \mid Z_1$, remove the edge between $X$ and $Y$.
3. For each $X$ and $Y$ which are still connected, if there is a third and a fourth variable $Z_1$ and $Z_2$ connected to $X$ or $Y$ such that $X \perp\!\!\!\perp Y \mid Z_1, Z_2$, remove the edge between $X$ and $Y$.
4. Iteratively increase the cardinality of the set of variables on which to condition.

To test for conditional independence we use the Hilbert-Schmidt conditional independence criterion $H_{XY|Z}$ for stochastic processes (Appendix A.1). The combination of the PC algorithm with a kernel-based dependence measure has been used in [13] and [12] where it is termed kPC. However, to our knowledge it has never been used in conjunction with a kernel-based measure of dependence for multidimensional stochastic processes.

Since the null distribution of the test statistics $H_{YX|Z}$ is not known, one possibility would be to use a permutation approach as in [12, Sec 2]. However the latter is not computationally efficient. We leave the development of a faster approach for future work, and adopt the approach [13] for this experiment. That is we use a threshold $\alpha$ and remove an edge if there is a $Z$ such that $H_{XY|Z} < \alpha$. We repeat 15 times the causal discovery procedure and use 30% of the runs to fix $\alpha$.

All experiments in Sec. 4 have been run on a P100 GPU to leverage an efficient dedicated CUDA implementation of the signature kernel.

## D Proofs

**Theorem 6.** *Given two stochastic processes $X, Y$*

$$\mathcal{D}_\mathcal{S}^2(X,Y) = 0 \iff \mathbb{P}_{X|\mathcal{F}_X} = \mathbb{P}_{Y|\mathcal{F}_Y}$$

*Furthermore*

$$\mathcal{D}_\mathcal{S}^2(X,Y) = 0 \implies \mathcal{D}_\mathcal{S}^1(X,Y) = 0$$

*but the converse is not generally true.*

*Proof.* First we note that by a standard result in signature kernel learning theory. e.g., [31], for $X \in \mathcal{X}(V)$ and every $t$, the mapping

$$\mathbb{P}_{X|\mathcal{F}_{X_t}} \mapsto \mu_{X|\mathcal{F}_{X_t}}^1 = \int k_\mathcal{S}(\cdot, x)\mathbb{P}_{X|\mathcal{F}_{X_t}}(dx)$$

is a homeomorphism (with respect to weak topology and Hilbert space topology); in particular, we have

$$\mathbb{P}_{X|\mathcal{F}_X} = \mathbb{P}_{Y|\mathcal{F}_Y} \iff \mathbb{P}_{\mu^1_{X|\mathcal{F}_X}} = \mathbb{P}_{\mu^1_{Y|\mathcal{F}_Y}}.$$

Then, using the same argument for $\mathbb{P}_{\mu^1_{X|\mathcal{F}_X}}$ and $\mathbb{P}_{\mu^1_{Y|\mathcal{F}_Y}}$, we can further deduce that

$$\mathbb{P}_{\mu^1_{X|\mathcal{F}_X}} = \mathbb{P}_{\mu^1_{Y|\mathcal{F}_Y}} \iff \int k_{\mathcal{S}}(\cdot, x)\mathbb{P}_{\mu^1_{X|\mathcal{F}_X}}(dx)(= \mu^2_X) = \int k_{\mathcal{S}}(\cdot, y)\mathbb{P}_{\mu^1_{Y|\mathcal{F}_Y}}(dy)(= \mu^2_Y).$$

Since by definition it holds that $\mathcal{D}^2_{\mathcal{S}}(X, Y) = \|\mu^2_X - \mu^2_Y\|_{\mathcal{H}^2(V)}$, we complete the proof of the first claim in this theorem.

For the second claim, it is easy to see that by definition $\mathbb{P}_{X|\mathcal{F}_X} = \mathbb{P}_{Y|\mathcal{F}_Y}$ ensures that $\mathbb{P}_X = \mathbb{P}_Y$, and therefore the implication that $\mathcal{D}^2_{\mathcal{S}}(X, Y) = 0 \implies \mathcal{D}^1_{\mathcal{S}}(X, Y) = 0$ follows immediately from the fact that $\mathbb{P}_X = \mathbb{P}_Y \iff \mathcal{D}^1_{\mathcal{S}}(X, Y) = 0$. Moreover, we refer readers to [26, Example 3.1] for a simple example which shows that there exist processes $X$ and $Y$ with $\mathcal{D}^1_{\mathcal{S}}(X, Y) = 0$ but $\mathcal{D}^2_{\mathcal{S}}(X, Y) > 0$. $\square$

**Theorem 7.** $\widehat{\mathcal{D}}^2_{\mathcal{S}}(X, Y)$ *is a consistent estimator for the $2^{nd}$ order MMD, i.e.*

$$|\widehat{\mathcal{D}}^2_{\mathcal{S}}(X, Y) - \mathcal{D}^2_{\mathcal{S}}(X, Y)| \overset{p}{\to} 0 \quad as \ m, n \to \infty \tag{39}$$

*with $\{x^i\}_{i=1}^m \sim X$, $\{y^i\}_{i=1}^n \sim Y$ and where convergence is in probability.*

*Proof.* Recall that given $m$ independent sample paths $\{x^i\}_{i=1}^m \sim X$, we can use the estimator in appendix A.2 to approximate sample paths $\{\widetilde{x}^i\}_{i=1}^m$ from the $1^{st}$ order predictive KME $\mu^1_{X|\mathcal{F}_X}$. Hence, it suffices to prove the following claim.

**Claim:** consider $m$ independent sample paths $\{\widetilde{x}^i\}_{i=1}^m \sim \mu^1_{X|\mathcal{F}_X}$. Then, the estimator given by $\widehat{\mu}^2_X = \frac{1}{m}\sum_{i=1}^m k_{\mathcal{S}}(\widetilde{x}^i, \cdot)$ is consistent for the $2^{nd}$ order predictive KME $\mu^2_X$, i.e.

$$\left\|\widehat{\mu}^2_X - \mu^2_X\right\|^2_{\mathcal{H}^2_{\mathcal{S}}(V)} \overset{p}{\to} 0, \ as \ m \to \infty \tag{40}$$

By the triangular inequality

$$\left\|\widehat{\mu}^2_X - \mu^2_X\right\|^2_{\mathcal{H}^2_{\mathcal{S}}(V)} = \left\|\frac{1}{m}\sum_{i=1}^m k_{\mathcal{S}}(\widetilde{x}^i, \cdot) - \mathbb{E}[k_{\mathcal{S}}(\mu^1_{X|\mathcal{F}_X}, \cdot)]\right\|^2_{\mathcal{H}^2_{\mathcal{S}}(V)} \tag{41}$$

$$\leq \left\|\frac{1}{m}\sum_{i=1}^m k_{\mathcal{S}}(\widetilde{x}^i, \cdot) - \mathbb{E}[k_{\mathcal{S}}(\widehat{\mu}^1_{X|\mathcal{F}_X}, \cdot)]\right\|^2_{\mathcal{H}^2_{\mathcal{S}}(V)} \tag{42}$$

$$+ \left\|\mathbb{E}[k_{\mathcal{S}}(\widehat{\mu}^1_{X|\mathcal{F}_X}, \cdot)] - \mathbb{E}[k_{\mathcal{S}}(\mu^1_{X|\mathcal{F}_X}, \cdot)]\right\|^2_{\mathcal{H}^2_{\mathcal{S}}(V)} \tag{43}$$

The term in (42) converges to 0 as $m \to \infty$ by the weak law of large numbers. Therefore, it remains to show that

$$\left\|\mathbb{E}_X[k_{\mathcal{S}}(\widehat{\mu}^1_{X|\mathcal{F}_X}, \cdot)] - \mathbb{E}_X[k_{\mathcal{S}}(\mu^1_{X|\mathcal{F}_X}, \cdot)]\right\|_{\mathcal{H}^2_{\mathcal{S}}(V)} \overset{p}{\to} 0, \ as \ m \to \infty \tag{44}$$

First note the following upper bound

$$\left\|\mathbb{E}_X[k_{\mathcal{S}}(\widehat{\mu}^1_{X|\mathcal{F}_X}, \cdot)] - \mathbb{E}_X[k_{\mathcal{S}}(\mu^1_{X|\mathcal{F}_X}, \cdot)]\right\|_{\mathcal{H}^2_{\mathcal{S}}(V)} \leq \mathbb{E}_X \left\|k_{\mathcal{S}}(\widehat{\mu}^1_{X|\mathcal{F}_X}, \cdot) - k_{\mathcal{S}}(\mu^1_{X|\mathcal{F}_X}, \cdot)\right\|_{\mathcal{H}^2_{\mathcal{S}}(V)}$$

We will show convergence of the right-hand-side. By [54, Theorem 3.4], for every $t = 1, \ldots, T$

$$\mathbb{E}_{X|\mathcal{F}_{X_t}} \left\|\widehat{\mu}^1_{X|\mathcal{F}_{X_t}} - \mu^1_{X|\mathcal{F}_{X_t}}\right\|^2_{\mathcal{H}_{\mathcal{S}}(V)} \overset{p}{\to} 0 \quad as \ m \to \infty \tag{45}$$

Now let us assume that the above convergences also hold almost surely for every $t = 1, \ldots, T$. Then by the Egorov's theorem, for any $\delta > 0$, there is a subset $\Omega_\delta$ with $\mathbb{P}(\Omega_\delta) > 1 - \delta$ and the above

convergence (45) holds uniformly on $\Omega_\delta$. This implies that on $\Omega_\delta$ for every $\varepsilon > 0$ there is an $N(\varepsilon)$ such that for all $m \geq N(\varepsilon)$ and all $t = 1, \ldots, T$, it holds that

$$\mathbb{E}_{X|\mathcal{F}_{X_t}} \left\| \widehat{\mu}^1_{X|\mathcal{F}_{X_t}} - \mu^1_{X|\mathcal{F}_{X_t}} \right\|^2_{\mathcal{H}_{\mathcal{S}}(V)} \leq \varepsilon^2 \tag{46}$$

From this estimate we immediately obtain by the triangle inequality on $\Omega_\delta$

$$\mathbb{E}_{X|\mathcal{F}_{X_t}} \left\| \widehat{\mu}^1_{X|\mathcal{F}_{X_t}} \right\|^2_{\mathcal{H}_{\mathcal{S}}(V)} \leq 2\mathbb{E}_{X|\mathcal{F}_{X_t}} \left\| \mu^1_{X|\mathcal{F}_{X_t}} \right\|^2_{\mathcal{H}_{\mathcal{S}}(V)} + 2\varepsilon^2, \tag{47}$$

and, by the Chebyshev's inequality, on $\Omega_\delta$, $\forall t = 1, \ldots, T$, $\forall m \geq N(\varepsilon)$

$$\mathbb{P}_{X|\mathcal{F}_{X_t}} \left[ \left\| \widehat{\mu}^1_{X|\mathcal{F}_{X_t}} - \mu^1_{X|\mathcal{F}_{X_t}} \right\|_{\mathcal{H}_{\mathcal{S}}(V)} > \sqrt{\varepsilon} \right] \leq \frac{1}{\varepsilon} \mathbb{E}_{X|\mathcal{F}_{X_t}} \left\| \widehat{\mu}^1_{X|\mathcal{F}_{X_t}} - \mu^1_{X|\mathcal{F}_{X_t}} \right\|^2_{\mathcal{H}_{\mathcal{S}}(V)} \leq \frac{1}{\varepsilon}\varepsilon^2 = \varepsilon,$$

which implies that on $\Omega_\delta$, $\forall t = 1, \ldots, T$, the sequence $\widehat{\mu}^1_{X|\mathcal{F}_{X_t}}$ converges to $\mu^1_{X|\mathcal{F}_{X_t}}$ in probability with respect to $\mathbb{P}_X$. By a standard result in rough path theory [30] there exists a universal constant $\beta \in \mathbb{R}$ such that

$$\left\| k_{\mathcal{S}}(\widehat{\mu}^1_{X|\mathcal{F}_X}, \cdot) \right\|_{\mathcal{H}^2_{\mathcal{S}}(V)} \leq \beta \left\| \widehat{\mu}^1_{X|\mathcal{F}_X} \right\|^{1-\mathrm{var}}_{\mathcal{H}_{\mathcal{S}}(V)} \tag{48}$$

where $\|\cdot\|^{1-\mathrm{var}}_{\mathcal{H}_{\mathcal{S}}(V)}$ denotes the total variation norm of paths taking values in $\mathcal{H}_{\mathcal{S}}(V)$. Since we are in a finite discrete time setup, it is easy to see that

$$\left\| \widehat{\mu}^1_{X|\mathcal{F}_X} \right\|^{1-\mathrm{var}}_{\mathcal{H}_{\mathcal{S}}(V)} \leq C(T) \sum_{t=1}^{T} \left\| \widehat{\mu}^1_{X|\mathcal{F}_{X_t}} \right\|_{\mathcal{H}_{\mathcal{S}}(V)} \tag{49}$$

Hence, combining all above arguments, we can conclude that on $\Omega_\delta$ and for all $m \geq N(\varepsilon)$,

$$\mathbb{E}_X \left\| k_{\mathcal{S}}(\widehat{\mu}^1_{X|\mathcal{F}_X}, \cdot) \right\|^2_{\mathcal{H}^2_{\mathcal{S}}(V)} \leq \beta^2 \mathbb{E}_X \left\| \widehat{\mu}^1_{X|\mathcal{F}_X} \right\|^2_{1-\mathrm{var}; \mathcal{H}_{\mathcal{S}}(V)} \tag{50}$$

$$\leq \beta^2 C(T) \sum_{t=1}^{T} \mathbb{E}_X \left\| \widehat{\mu}^1_{X|\mathcal{F}_{X_t}} \right\|^2_{\mathcal{H}_{\mathcal{S}}(V)} \tag{51}$$

$$\leq \beta^2 C(T) \left( 2 \sum_{t=1}^{T} \mathbb{E}_X \left\| \mu^1_{X|\mathcal{F}_{X_t}} \right\|^2_{\mathcal{H}_{\mathcal{S}}(V)} + 2T\varepsilon^2 \right) \leq C < \infty \tag{52}$$

where in the last line we used (47). As a result, we obtain that on $\Omega_\delta$,

$$\sup_{m \geq N(\varepsilon)} \mathbb{E}_X \left\| k_{\mathcal{S}}(\widehat{\mu}^1_{X|\mathcal{F}_X}, \cdot) - k_{\mathcal{S}}(\mu^1_{X|\mathcal{F}_X}, \cdot) \right\|^2_{\mathcal{H}^2_{\mathcal{S}}(V)} < \infty \tag{53}$$

which in turn implies, by the de la Vallée–Poussin theorem, that on $\Omega_\delta$, the sequence $\left\| k_{\mathcal{S}}(\widehat{\mu}^1_{X|\mathcal{F}_X}, \cdot) - k_{\mathcal{S}}(\mu^1_{X|\mathcal{F}_X}, \cdot) \right\|^2_{\mathcal{H}^2_{\mathcal{S}}(V)}$, $m \geq N(\varepsilon)$ is uniformly integrable for $\mathbb{P}_X$. Then recalling that we have shown that $\widehat{\mu}^1_{X|\mathcal{F}_X}$ converges to $\mu^1_{X|\mathcal{F}_X}$ in probability with respect to $\mathbb{P}_X$ as $m \to \infty$, a standard result in probability theory ensures that (thanks to the uniform integrability of the sequence and the continuity of the kernel $k_{\mathcal{S}}$ which ensures that $\left\| k_{\mathcal{S}}(\widehat{\mu}^1_{X|\mathcal{F}_X}, \cdot) - k_{\mathcal{S}}(\mu^1_{X|\mathcal{F}_X}, \cdot) \right\|_{\mathcal{H}^2_{\mathcal{S}}(V)} \to 0$ in probability for $\mathbb{P}_X$) on $\Omega_\delta$, $\mathbb{E}_X \left\| k_{\mathcal{S}}(\widehat{\mu}^1_{X|\mathcal{F}_X}, \cdot) - k_{\mathcal{S}}(\mu^1_{X|\mathcal{F}_X}, \cdot) \right\|_{\mathcal{H}^2_{\mathcal{S}}(V)} \to 0$, as $m \to \infty$, which implies that $\left\| \mathbb{E}_X[k_{\mathcal{S}}(\widehat{\mu}^1_{X|\mathcal{F}_X}, \cdot)] - \mathbb{E}_X[k_{\mathcal{S}}(\mu^1_{X|\mathcal{F}_X}, \cdot)] \right\|_{\mathcal{H}^2_{\mathcal{S}}(V)} \to 0$, as $m \to \infty$, on $\Omega_\delta$. Clearly, as $\delta$ was arbitrary, we have $\mathbb{E}_{X|\mathcal{F}_{X_t}} \left\| \widehat{\mu}^1_{X|\mathcal{F}_{X_t}} - \mu^1_{X|\mathcal{F}_{X_t}} \right\|^2_{\mathcal{H}_{\mathcal{S}}(V)} \to 0$ as $m \to \infty$ a.s. Finally, note that the above result holds true for any subsequence of $(\widehat{\mu}^1_{X|\mathcal{F}_{X_t}})_{m \geq 1}$ (because every sequence converging in probability has a subsequence converging almost surely), which proves the desired result (40). $\square$

**Theorem 8.** *Given two stochastic processes $X, Y$*

$$\mathcal{D}_{\mathcal{S}}^n(X, Y) = 0 \implies \mathcal{D}_{\mathcal{S}}^k(X, Y) = 0 \quad \text{for any } 1 < k < n \tag{54}$$

*but the converse is not generally true.*

*Proof.* Let $X \in \mathcal{X}(V)$, then we denote $\mathbb{P}_{X|\mathcal{F}_{X_t}} =: X_t^{(1)}$. Then we continue this procedure and define $X_t^{(n)} := \mathbb{P}_{X^{n-1}|\mathcal{F}_{X_t}}$ (it is called the rank $n$ prediction process in [29]). Now we can apply the same argument as in the proof of Theorem 6 together with an induction procedure easily prove that

$$\mathcal{D}_{\mathcal{S}}^n(X, Y) = 0 \iff \mathbb{P}_{X^{(n-1)}} = \mathbb{P}_{Y^{(n-1)}} \tag{55}$$

for all $n > 1$. From the definition of these processes $X^{(n)}$ and $Y^{(n)}$ we can immediately see that $\mathbb{P}_{X^{(n)}} = \mathbb{P}_{Y^{(n)}}$ ensures that $\mathbb{P}_{X^{(k)}} = \mathbb{P}_{Y^{(k)}}$ for all $k < n$, which yields the desired result. We refer readers to [26, Example 3.2] for examples which illustrate that for each $n$ there exist processes $X$ and $Y$ with $\mathcal{D}_{\mathcal{S}}^n(X, Y) = 0$ (equivalently, $\mathbb{P}_{X^{(n-1)}} = \mathbb{P}_{Y^{(n-1)}}$) but $\mathcal{D}_{\mathcal{S}}^{n+1}(X, Y) > 0$ (equivalently, $\mathbb{P}_{X^{(n)}} \neq \mathbb{P}_{Y^{(n)}}$). $\square$

**Remark.** Using terminologies from [26] and [29], $\mathbb{P}_{X^{(n)}} = \mathbb{P}_{Y^{(n)}}$ means that processes $X$ and $Y$ have the same adapted distribution up to rank $n$. Therefore Thm. 6 and 8 tell us that $\mathcal{D}_{\mathcal{S}}^n(X, Y) = 0$ if and only if they ave the same adapted distribution up to rank $n$. Moreover, using the partial isometry between RKHS generated by $k_{\mathcal{S}}$ and tensor algebra space, see e.g. [31, Theorem E.2], one can use an induction argument to verify that $\mathcal{D}_{\mathcal{S}}^n$ coincides with the metric $d_{n-1}$ defined in [29, Definition 14], and therefore by [29, Theorem 4] we can obtain a stronger result that $\mathcal{D}_{\mathcal{S}}^n$ actually metrizes the so–called rank $n-1$ adapted topology (see [29, Definition 5], [26, Definition 2.25]). For more details regarding adapted topologies we refer to [29].

**Theorem 9.** *Let $f : \mathbb{R} \to \mathbb{R}$ be a globally analytic function with non-negative coefficients. Define the family of kernels $K_{\mathcal{S}}^n : \mathcal{P}(\mathcal{X}(V)) \times \mathcal{P}(\mathcal{X}(V)) \to \mathbb{R}$ as follows*

$$K_{\mathcal{S}}^n(X, Y) = f(\mathcal{D}_{\mathcal{S}}^n(X, Y)), \quad n \in \mathbb{N}_{\geq 1} \tag{56}$$

*Then the RKHS associated to $K_{\mathcal{S}}^n$ is dense in the space of functions from $\mathcal{P}(\mathcal{X}(V))$ to $\mathbb{R}$ which are continuous with respect to the $k^{th}$ order MMD for any $1 \leq k \leq n$.*

*Proof.* By [55, Thm. 2.2] if $K$ is a compact metric space and $H$ is a separable Hilbert space such that there exists a continuous (w.r.t. a topology $\tau$ on $K$) and injective map $\rho : K \to H$, then for any globally analytic function with non-negative coefficients $f : \mathbb{R} \to \mathbb{R}$ the kernel $k : K \times K \to \mathbb{R}$ given by

$$k(z, z') = f\left(\|\rho(z) - \rho(z')\|_H\right) \tag{57}$$

is universal in the sense that its RKHS is $\tau$-dense in the space of $\tau$-continuous functions from $K$ to $\mathbb{R}$. By assumption, $\mathcal{X}(V)$ is a $\mathcal{D}_{\mathcal{S}}^1$-compact metric space, therefore by Thm. 8 it is also $\mathcal{D}_{\mathcal{S}}^n$-compact for every $n \geq 1$. Hence, by [56, Thm. 10.2] the set of stochastic processes $\mathcal{P}(\mathcal{X}(V))$ is also $\mathcal{D}_{\mathcal{S}}^n$-compact. For showing that $\rho : X \mapsto \mu_X^n$ is injective and continous with respect to $\mathcal{D}_{\mathcal{S}}^n$ we refer readers to [29, Proposition 4], one only needs to verify that $\mu_X^n$ corresponds to the mapping $\bar{S}_n$ used in the proof of [29, Proposition 4] by the definition of $\mu_X^n$ and the fact that $\mathcal{D}_{\mathcal{S}}^n$ metrizes the rank $n-1$ adapted topology (cf. the above remark). Furthermore $\mathcal{H}_{\mathcal{S}}^n(V)$ can be shown by induction to be a Hilbert space with a countable basis, hence it is separable. Setting $K = \mathcal{P}(\mathcal{X}(V)), H = \mathcal{H}_{\mathcal{S}}^n(V)$ and $\rho : X \mapsto \mu_X^n$ concludes the proof. $\square$