# OpenReview forum: "Higher Order Kernel Mean Embeddings to Capture Filtrations of Stochastic Processes"
_NeurIPS.cc/2021/Conference — NeurIPS 2021 Poster_

### Official Review · Reviewer_V4uV · 2021-07-16

**Rating:** 7
**Confidence:** 4

**Summary:**

This paper aims to construct a family of kernel mean embeddings that capture filtration information from stochastic processes. This is achieved through the use of the signature kernel, a characteristic kernel defined on pairs of paths that can be evaluated by solving a hyperbolic PDE, and by forming conditional kernel mean embeddings of the stochastic process conditioned on its own filtration up to some time in the past.

The resulting conditional kernel mean operator itself is a stochastic process, and thus this construction can be repeated to produce higher ordered conditional kernel mean embeddings of the original stochastic process. The corresponding MMD defined from the conditional kernel mean embeddings at each order can then be used to measure distances between two stochastic processes at each order.

The first major contribution shows that higher ordered MMDs are stronger discrepancy measures than lower ordered MMDs (Theorem 2 and 4). They can then be used to perform a kernel two-sample test between paths from two stochastic processes. In particular, the paper focuses on using the second order MMD and derive consistent estimators for it (Theorem 3).

The second major contribution applies these constructions for distribution regression on stochastic processes, where the input is a collection of sample paths and the output is a vector of scalars. This is done by embedding the sampled paths from the stochastic process into a kernel mean embedding through the signature kernel, then learning the mapping between this kernel mean embedding to the vector of scalar outputs. The authors pay strong attention to the application in financial markets where the functions of interest are often discontinuous with respect to the first order mean embedding, and show they can still be learned by proving that the RKHS induced in higher orders is dense in the space of functions of interest (Theorem 5).

**Limitations And Societal Impact:**

The work is motivated by applications in financial markets. However, its contributions are general in nature in terms of the technique and not limited to financial markets.

The paper can be improved by highlighting potential wider impacts in being able to capture filtration information through kernel mean embeddings. I believe these concepts are more general and can be applied to settings other than kernel two-sample tests and distribution regression.

**Main Review:**

Overall, the contributions of the paper is well motivated, well articulated, and technically interesting. The developments presented in the paper is clear and well ordered. It is clear why the signature kernel was chosen and the motivation behind constructing conditional kernel mean embeddings of stochastic processes conditioned on their filtrations. The subsequent developments in constructing higher ordered kernel mean embeddings and their applications to kernel two-sample testing and distribution regression is also easily to follow.

To me, the major novelty presented in this paper is the notion of recursively constructing the conditional kernel mean embedding of stochastic processes conditioned on its own filtration into high ordered versions of the original, by making use of the fact that the resulting conditional kernel mean embedding (or more exactly the conditional kernel mean operator, with the conditioned variable left as a stochastic process not as a realised path) is itself a stochastic processes. Combined with the fact that discrepancies become stronger at higher orders, as measured by the MMDs of the respective order, this is a very general notion for learning tasks and relationships involving stochastic processes with an ordered nature to its entirety, without losing information from filtration.

The two example applications presented in this paper, kernel two-sample testing and distribution regression, are rather straight forward in consequence by applying what we know about these applications of KMEs in general to this specific setting where KMEs are constructed as previously described. However, they are well developed nonetheless.

This paper can be further improved by some discussion on how the kernel hyperparameters can be learned or selected. Especially at higher orders, hyperparameter learning does not seem trivial and presumably performance can be quite sensitive to the kernel hyperparameters unless we can argue otherwise.

Overall, the notions introduced in this paper can be quite valuable to the wider community in extending adoption of KME based methods to also tasks involving stochastic processes where filtration information is important to capture. I would tend towards acceptance.

**Time Spent Reviewing:**

3

---

> ### Author Response · Authors · 2021-08-09
> **Official comment for reviewer V4uV**
>
> We thank reviewer V4uV for the useful feedback. Please find our responses below.
>
> **This paper can be further improved by some discussion on how the kernel hyperparameters can be learned or selected. Especially at higher orders, hyperparameter learning does not seem trivial and presumably performance can be quite sensitive to the kernel hyperparameters unless we can argue otherwise.**
>
> Regarding  the  choice  of  kernel  hyperparameters, in the setting of two-sample tests, we can use various hyperparameter selection methods which have been proposed in the kernel literature, including approaches aiming at maximizing the test power using the signal-to-noise-ratio as an objective [1, 2, 3].  In the distribution regression setting, we made use of a cross validation approach.  Developing hyperparameter tuning methodologies for higher order kernel mean embeddings is an interesting future work direction and we will note that [4, 5] are certainly a good starting point for such an investigation.
>
> **The paper can be improved by highlighting potential wider impacts in being able to capture filtration information through kernel mean embeddings. I believe these concepts are more general and can be applied to settings other than kernel two-sample tests and distribution regression.**
>
> Indeed, we hope to see further applications of the higher order kernel mean embeddings. We propose to add the following remark to highlight potential wider impacts: Higher  order  kernel mean  embeddings  have  the  potential  to  be  used  beyond  two-sample  tests  and  distribution  regression.   For  example  [6]  recently  investigated the use of the $1^{st}$ order MMD to derive an approximate Bayesian computation (ABC) algorithm for irregular time series.  Another idea that is currently being investigated  is  using  higher  order  MMDs  as  discriminators  in  autoregressive generative models for time series, where conditioning the future trajectories on past observations is key.
>
> **References**
>
> [1] Arthur Gretton, Dino Sejdinovic, Heiko Strathmann, Sivaraman Balakrish-nan, Massimiliano Pontil, Kenji Fukumizu, and Bharath K Sriperumbudur. *Optimal kernel choice for large-scale two-sample tests. In Advances in neural information processing systems*, pages 1205–1213. Citeseer, 2012.
>
> [2] Danica J. Sutherland, Hsiao-Yu Tung, Heiko Strathmann, Soumyajit De, Aaditya  Ramdas,  Alexander  J.  Smola,  and  Arthur  Gretton.   *Generative models and model criticism via optimized maximum mean discrepancy*.  In 5th International Conference on Learning Representations, ICLR, 2017.
>
> [3] Feng  Liu,  Wenkai  Xu,  Jie  Lu,  Guangquan  Zhang,  Arthur  Gretton,  and Danica J Sutherland. *Learning deep kernels for non-parametric two-sample tests*. In International Conference on Machine Learning, pages 6316–6326. PMLR, 2020.
>
> [4] Seth Flaxman, Dino Sejdinovic, John P Cunningham, and Sarah Filippi. *Bayesian learning of kernel embeddings*. In Proceedings of the Thirty-Second Conference on Uncertainty in Artificial Intelligence, pages 182–191, 2016.
>
> [5] Kelvin Hsu, Richard Nock, and Fabio Ramos. *Hyperparameter learning for conditional kernel mean embeddings with rademacher complexity bounds. In Joint  European  Conference  on  Machine  Learning  and  Knowledge  Discovery in Databases*, pages 227–242. Springer, 2018.
>
> [6] Joel   Dyer,   Patrick   Cannon,   and   Sebastian   M   Schmon. *Approximate   bayesian   computation   with   path   signatures*. arXiv   preprint arXiv:2106.12555, 2021.

---

> > ### Comment · Reviewer_V4uV · 2021-08-26
> > **Thank you for your response**
> >
> > Thank you for your response. I would suggest the authors to include these discussions on future work and extensions regarding hyperparameter learning and other potential areas of application. I still maintain that hyperparameter learning is not trivial in your setting of higher ordered KMEs, in order to make this easily applicable. Nevertheless, I will keep my original score.

---

### Official Review · Reviewer_zRy1 · 2021-07-16

**Rating:** 7
**Confidence:** 3

**Summary:**

The paper proposes (higher order) kernel mean embeddings (KMEs) for stochastic processes -- the novelty is in considering the filtration induced by the process rather than the sample path, which allows us to consider historical dependence.


**Main Review:**

My overall impression of the paper is positive.
The paper introduces an interesting way of summarising stochastic processes, reflecting their history information (given in the form of filtration).
As demonstrated by the authors with examples, the maximum mean discrepancy (MMD) defined by the new embedding allows us to characterise differences between processes, which are not captured by the KME defined by the law of the path-valued random variable corresponding to the process. The latter approach is, to my knowledge, common in areas like functional data analysis, whereas we often need to exploit past information (dependence).
The example applications make the KMEs look promising, although some of the applications are not fully established (e.g., a two-sample test is actually not proposed); the proposed approach would lead to new applications of the MMD (or the KMEs) for sequential data, as done for the original MMD.

## Comments
* Higher-order KMEs are defined in terms of conditional distributions. Is the space of the recursive RKHS regular enough so that regular conditional distributions are defined? It would be great this point (e.g., with kernel's continuity and separability of $V$ )
* How does the process in (14) satisfy the assumption on $\mathcal{X}$ given in the first paragraph of Section 2?
* The conditional independence testing with continuous conditioning variables is known as hard [Shah and Peters, 2020]  Although the experiment result in Section 4.3 looks reasonable, it might need to be replaced with some other results.


## Minor comments:
* L133: $\mathcal{H}(\mathbb{R}^d)$ -> $\mathcal{H}(V)$
* Some equations do not have punctuation marks, such as the period.
* The Markov process figure (Figure 2) is missing the X-axis; adding time indices would help.
* Numbering to equations would be better if it is limited to the ones that are actually referenced (I assume the authors did it for the reviewing purpose).
* Is mentioning Eq. (12) necessary? On the RHS, the CME acts on the kernel; how is this defined? As done in Theorem 4 in [Song, Huang, Smola, and Fukumizu, 2009]? The operator definition is less general than the Bochner integral definition.

References:
* Rajen D. Shah, Jonas Peters "The hardness of conditional independence testing and the generalised covariance measure," The Annals of Statistics, Ann. Statist. 48(3), 1514-1538, (June 2020)

**Time Spent Reviewing:**

10 hours

---

> ### Author Response · Authors · 2021-08-09
> **Official comment to reviewer zRy1**
>
> We thank reviewer zRy1 for the useful feedback. Please find our responses below.
>
> **Higher-order KMEs are defined in terms of conditional distributions.  Is the space of the recursive RKHS regular enough so that regular conditional distributions  are  defined?  It  would  be  great  this  point  (e.g.,  with  kernel’s continuity and separability of V)**
>
> Yes, since $V$ is a Polish space (i.e.,  a separable,  complete metric space) and  signature  maps  are  always  continuous,  in  view  of  [1,  Lemma  4.33] one  can  easily  check  that  all  RKHSs  appearing in  the  present  paper  are seprable Hilbert spaces by an induction argument and therefore all regular conditional distributions are well–defined.
>
> **How does the process in (14) satisfy the assumption on given in the first paragraph of Section 2?**
>
> First note that for each $t$ we have (cf. Eq.(13))
>
> $\mu^1_{X|F_{X_t}} = \int_{x \in X(V)} k_S(\cdot,x)P_{X|F_{X_t}}(dx)$
>
> is the Bochner integral of $k_S(\cdot,x)$ with respect to the probability measure $P_{X|F_{X_t}}(dx)$. Since we implicitly assume that $V \subset R^d$ is a compact set, the pathspace $X(V)$ is compact for the product topology; then as $x \mapsto k_{S}(\cdot,x)$ is continuous, the set $K:=$ {$k_S(\cdot,x): x \in X(V)$} is compact as continuous image of a compact set, and therefore its Bochner integral $\mu^1_{X|F_{X_t}}$ takes values in the closed convex hull of $K$, which is again a compact set in the RKHS $H_{S}(V)$. Consequently the path $t\mapsto\mu^1_{X|F_{X_t}}$ belongs to a compact subset of $X(H_{S}(V))$ (for the product topology), which satisfies the assumptions introduced in Section 2.
>
> **The conditional independence testing with continuous conditioning variables  is  known  as  hard  [Shah  and  Peters,  2020].  Although  the  experiment result  in  Section  4.3  looks  reasonable,  it  might  need  to  be  replaced  with some other results**
>
> As outlined in $\ell$ 174-175 and Appendix A, the formalism of covariance and cross-covariance operators on RKHSs is essential for the construction of higher order KMEs/MMDs. The observation we make is that computing these objects allows one to build a criterion for conditional independence of stochastic processes $X \perp Y \mid Z \iff H_{YX|Z}=0$ and then use the kernel PC algorithm [2] for causal discovery in Sec 4.3.
>
> However,  we  recognize  that  it  is  important  to  mention  that  for  finite datasets conditional independence testing is hard without additional assumptions,  as  discussed  in  [Shah  and  Peters,  2020]  and  [3],  which  we propose to cite.  We are not sure what are the ”some other results” the reviewer is referring to, but we are of course happy to have further discussions about this comment.
>
> **Some equations do not have punctuation marks, such as the period.  The Markov process figure (Figure 2) is missing the X-axis; adding time indices would  help.   Numbering  to  equations  would  be  better  if  it  is  limited  to the  ones  that  are  actually  referenced  (I  assume  the  authors  did  it  for  the reviewing purpose).  Is mentioning Eq.  (12) necessary?**
>
> Thanks for pointing out the issues of punctuation, notation ($H(R^d)$), missing axis in Fig.  2 and unnecessary equation numbering; we will address them in the final version.  In addition, eq.  (12) is indeed not necessary and so will be removed.
>
> **References**
>
> [1] Andreas Christmann  and  Ingo  Steinwart. *Support  Vector  Machines*. Springer verlag, 2008
>
> [2] Robert E Tillman, Arthur Gretton, and Peter Spirtes.  *Nonlinear directedacyclic  structure  learning  with  weakly  additive  noise  models*.  In NIPS , pages 1847–1855, 2009
>
> [3] Anton Rask Lundborg, Rajen D Shah, and Jonas Peters.  *Conditional independence testing in Hilbert spaces with applications to functional data analysis*. arXiv preprint arXiv:2101.07108, 2021

---

> > ### Comment · Reviewer_zRy1 · 2021-08-25
> > **Re: Official comment to reviewer zRy1**
> >
> > I appreciate the detailed response.
> > For the causal discovery application, I think citing the suggested papers, with discussion on the limitation, would be enough.
> >
> > I have minor questions.
> > What is the integrablity condition for a higher-order KME to exist? Can we express it in terms of the input path $x$? (It feels that as long as the path is regular enough, the total variation norm of higher-order signatures shouldn't be affected).
> > On a related note, the remark in the first sentence in L82, does this not require additional assumptions (square summable power series & completion)?

---

> > > ### Author Response · Authors · 2021-08-25
> > > **Reply to reviewer zRy1**
> > >
> > > We thank the reviewer for the two remarks.
> > >
> > > Firstly, as mentioned in L70-L72, all paths considered in the present paper are piecewise linear. Consequently, all sample paths from higher order predictive KMEs are also piecewise linear and their KMEs are always well defined. Such a nice property will not hold anymore if we consider more generic continuous sample paths, because such regularity of sample paths from the corresponding higher order predictive KMEs might break as noted in Remark 1 in  [1]. The study of how the regularity changes by taking higher order kernel mean embeddings is an interesting direction for future work.
> > >
> > > For the second question: you are correct, we do need to take completion of square summable power series, please see [2] or Def. 3.1 in [3] for more details. In the present paper we also referred the reader to [2] for the rigorous definition of that space.
> > >
> > > **References**
> > >
> > > [1] Patric Bonnier, Chong Liu, and Harald Oberhauser. *Adapted topologies and higher rank signatures*. arXiv:2005.08897, 2020.
> > >
> > > [2] Franz J Király and Harald Oberhauser. *Kernels for sequentially ordered data*. Journal of Machine Learning Research, 2019.
> > >
> > > [3] Ilya Chevyrev and Harald Oberhauser. *Signature moments to characterize laws of stochastic processes*. arXiv:1810.10971, 2018.

---

> > > > ### Comment · Reviewer_zRy1 · 2021-08-25
> > > > **Thanks message**
> > > >
> > > > Thank you for taking the time to respond in detail to my questions. I appreciate the response.

---

### Official Review · Reviewer_SVrX · 2021-07-16

**Rating:** 7
**Confidence:** 3

**Summary:**

The paper proposes a generalized approach to kernel mean embeddings – higher-order mean embeddings --  and new two-sample test based on the newly proposed kernel which can be applied to stochastic processes and in particular is designed to capture filtration information in stochastic processes. Various theoretical results are derived, including consistency.


**Limitations And Societal Impact:**

I do not see obvious negative societal impact.

**Main Review:**

The paper proposes an interesting new approach which can be used, in particular, in quantitative finance. The paper also presents an application to causal discovery. While I found the approach and motivation relevant, the applications described in the paper appeared to me rather limited. I believe more extensive experiments have to be performed to fully understand the advantages of the proposed method. The current experiments have to be explained in more detail and clearly motivated.

Some detailed comments:
-	I found it a bit hard to follow at times, perhaps describing the outline of the paper in the end of the introduction with the main contributions could be helpful;
-	Finance application could use more description;  it is really not a classical finance model and since the finance audience is also not targeted audience of the paper it would be good to provide more extensive intuition about what it is used for, what are particular challenges etc.
-	Related to the previous point: the authors do mention that the correlation parameter \rho is of particular interest; in the simulation studies the authors draw values for \rho from the interval [-0.4;0.4]. Is this reasonable? I would have expected a higher (positive and negative) correlation be possible as well.
-	$h$ is not introduced (I suppose it is related to the frequency of sampling of the data)? And the choice of this parameter for the simulation study is not motivated.
-	The choice of the kernels for the comparison seems to be not ideal; in particular I would indeed expect the RBF kernel to be not appropriate since it is high-frequency financial data (so it is unlikely to be smooth).
-	Considering different numbers of rough volatility trajectories also could strengthen this application (see [6]).  In particular, if the advantage is the same over different numbers of trajectories would be of interest.
-	How is the choice of $h$ motivated in the simulation studies?
-	In the causal discovery application I do not see a competitive method to which we could compare the proposed method. How does it perform in comparison to $K_{S}^{1}$, for example? Comparison to the state-of-the-art conditional independence test perhaps could highlight the strength of the approach.

UPD: Most of my original concerns were about empirical applications and the choices the authors made for the kernels/parameters range etc. During the rebuttal and discussion, the authors clarified the things I missed and provided additional results which are in line with the literature for the parameter choice of the financial model. These results confirmed earlier findings. Taking into account the addressed concerns and novelty of the proposed method I change my score to 7 and recommend the paper for acceptance.

**Time Spent Reviewing:**

3,5

---

> ### Author Response · Authors · 2021-08-09
> **Official comment to reviewer SVrX**
>
> We thank reviewer SVrX for the useful feedback. Please find our responses below.
>
> **The applications described in the paper appeared to me rather limited.  I believe more extensive experiments have to be performed to fully understand the advantages of the proposed method.  The current experiments have to be explained in more detail and clearly motivated.**
>
> The main focus of the paper is to introduce a new family of KMEs (establishing theoretical properties in Thms. 2. 4. and 5. for downstream learning tasks) that might be used by the wider community for tasks involving stochastic processes where  filtration  information  is  important  to  capture.  To  this  aim  we  supplement the theoretical foundations with example applications which according toreviewer V4uV are “straightforward” but “well developed”.
>
> **I found it a bit hard to follow at times, perhaps describing the outline of the paper in the end of the introduction with the main contributions could be helpful.**
>
> We will expand the outline of the paper in the end of the introduction $\ell$ 30-40.
>
> **Finance application could use more description; it is really not a classical finance model and since the finance audience is also not targeted audience of  the  paper  it  would  be  good  to  provide  more  extensive  intuition  about what it is used for, what are particular challenges etc.**
>
> We give a summary of the essentials of a rough Bergomi model in Sec 4.2, but we agree that further explanation about the financial details could be added in Appendix C. We propose to add the following paragraph to Appendix C:
>
> *Rough volatility models constitute a class of models that are empirically well-tailored to fit observed implied market volatilities in the context of option pricing for short maturity assets. The basic model for option pricingis  called  the  Black-Scholes  model  in  which  the  volatility  is  assumed  to be  constant.   Stochastic  volatility  models  are  extensions  of  the  Black-Scholes  model  to  the  case  where  the  volatility  is  itself  stochastic.   The main  shortcoming  of  such  stochastic  volatility  models  is  that  they  are able to capture the true steepness of the implied volatility smile close to maturity (see [1] for extra details).  This is where rough volatility models become useful. Among them, the rough Bergomi model introduced by [1], stood out for its ability to explain implied volatility and other phenomena related to European options.*
>
> **The authors do mention that the correlation parameter $\rho$ is of particular interest; in the simulation studies the authors draw values for $\rho$ from the interval $[−0.4; 0.4]$.   Is  this  reasonable?   I  would  have  expected  a  higher (positive and negative) correlation be possible as well.**
>
> We agree that higher correlations are possible, but experimentally we observed that this regime is more challenging than a wider interval.
>
> **$h$ is  not  introduced  (I  suppose  it  is  related  to  the  frequency  of  sampling of the data)?  And the choice of this parameter for the simulation study is not motivated.**
>
> In the calibration experiment $h$ is introduced at $\ell$ 248-249. Choosing $h \in (0,1)$ ensures that the model is non-Markovian, i.e. is a Rough Bergomi (rBergomi) model.  Besides, rBergomi models with $h \in (0,1/2)$ fit the observed volatility surface better than conventional Markovian stochastic volatility models [1].  We will add this remark in the final version of the paper.  In the option pricing experiment $h$ is introduced in $\ell$ 268-270.  We follow the experimental setup from [2].
>
> **The  choice  of  the  kernels  for  the  comparison  seems  to  be  not  ideal;  in particular  I  would  indeed  expect  the  RBF  kernel  to  be  not  appropriate since it is high-frequency financial data (so it is unlikely to be smooth).**
>
> We point out that we are not using the Gaussian kernel $k(x,y) = \exp(−\sigma^2||x−y||^2)$ to interpolate high-frequency financial data.  The RBF baseline refers to a distribution regression kernel in the form of eq. (27) with the MMD associated  to  the  Gaussian  kernel  also  used  in  [3].   This  is  detailed  in Appendix  C,  however  we  agree  that  we  should  make  this  clearer  in  the main paper.  Given, to the best of our knowledge, the lack of kernels on probability measures on paths, the RBF baseline is a reasonable choice.  In effect, as per [4, Thm.  5], the Gaussian kernel is characteristic for the infinite-dimensional inner product space of sequences vanishing at infinity.
>
> **In  the  causal  discovery  application  I  do  not  see  a  competitive  method to  which  we  could  compare  the  proposed  method.   How  does  it  performin  comparison  to $K^1_S$,  for  example?   Comparison  to  the  state-of-the-art conditional  independence  test  perhaps  could  highlight  the  strength  of  the approach.**
>
> We agree that adding competitive methods could highlight the strength of the approach.  However, we point out that the distribution regression kernel $K^1_S$ is  not  applicable  in  the  causal  discovery  setting. Instead we propose to add the RBF and Matern embeddings as baselines for comparison.
>
> **References**
>
> [1] Christian Bayer, Peter Friz, and Jim Gatheral. *Pricing under rough volatility*. Quantitative Finance, 16(6):887–904, 2016.
>
> [2] Calypso Herrera, Florian Krach, Pierre Ruyssen, and Josef Teichmann. *Optimal stopping via randomized neural networks*. arXiv preprint arXiv:2104.13669, 2021.
>
> [3] Maud Lemercier,  Cristopher Salvi,  Theodoros Damoulas,  Edwin Bonilla,and Terry Lyons.  *Distribution regression for sequential data*.  International Conference on Artificial Intelligence and Statistics, pages 3754–3762.PMLR, 2021.
>
> [4] Saeed Hayati, Kenji Fukumizu, and Afshin Parvardeh.  *Kernel mean embedding  of  probability  measures  and  its  applications  to  functional  data analysis*. arXiv preprint arXiv:2011.02315, 2020.

---

> > ### Comment · Reviewer_SVrX · 2021-08-24
> > **Reviewer reply**
> >
> > I thank the authors for the detailed reply and apologize for returning with questions and suggestions quite late. Most of the points have clarified my concerns. Nevertheless, I would like to still raise a couple of issues.
> >
> > **We agree that higher correlations are possible, but experimentally we observed that this regime is more challenging than a wider interval.**
> >
> > Do you have an idea why this is a more challenging regime? It's quite surprising as I would have expected a wider interval and higher correlations to be more difficult. In particular, in [1] \rho is set to -0.9 if I am not mistaken, and generally, high correlation (positive or negative) is quite realistic for financial data. In other papers with this model I looked at (for this model specifically), the correlation is assumed to be in [-1,1], or different regimes were considered (typically -0.9 included). Could you elaborate in what way you find \rho\in[-0.4;0.4] more challenging?
> >
> > A minor thing: when looking up the references, I noticed that the capitals are missing. For example bergomi in [38] -> Bergomi.
> >
> > I also noticed some reproducibility issues:
> >
> > When running Hypothesis_testing_on_filtrations.ipynb
> >
> > In file sigkernel.py line 6 is commented out (from cython_backend import sig_kernel_batch_varpar, sig_kernel_Gram_varpar). However, without it, I get an error ‘NameError: name 'sig_kernel_Gram_varpar' is not defined’. In any case, there seems to be no module cython_backend, or I am missing something. Do you have an idea what the problem might be?
> >
> > When running Higher_order_DR.ipynb
> >
> > The same issue appears in Higher_order_DR.ipynb (cell 14) (the root of the problem is in the same file – sigkenel.py).
> >
> > Thus, unfortunately, I could not reproduce/verify the results.
> >
> > Additional tip wrt to the code: GridSearchCV gave an error for me with n_jobs=-1 (seems to be a known problem online, but it might be worth flagging this one in the comments, n_jobs=1 should be working fine for everyone).

---

> > > ### Author Response · Authors · 2021-08-25
> > > **Reply to reviewer SVrX**
> > >
> > > We thank the reviewer for the additional feedback.
> > >
> > > Regarding the range for the parameter $\rho$ in the rough Bergomi model calibration, for completeness please find below the results for the experiment with $\rho$ in the suggested range $[-1,1]$.
> > >
> > > | Model      | Accuracy |
> > > | ----------- | ----------- |
> > > | RBF     |     87 %   |
> > > | Matern   |  87  %      |
> > > | $K^1_S$  | 91  %      |
> > > | $K^2_S$   | 93  %     |
> > >
> > > As you can see, all the models (except Matern) improve in accuracy, and that is why we preferred to report the results for the more challenging regime $[-0.4,0.4]$. However, we do not have a financial justification for explaining why this is the case, and therefore we propose to follow your advice and report these results instead of the previous ones.
> > >
> > > As regards the reproducibility issues you have encountered, to run the experiments a GPU is required (as mentioned in L137 in Appendix C). The original sigkernel code used throughout the experimental section also has a cython backend, which is significantly slower than the cuda alternative. This is why we only included the later in the provided code. To allow the user to run the code on a CPU we will uncomment line 6 and add the cython backend files.

---

> > > > ### Comment · Reviewer_SVrX · 2021-08-26
> > > > **Reply reviewer**
> > > >
> > > > That's great; thank you for following up and providing additional results. I think this is a more reasonable assumption and is in line with the literature, so I would indeed recommend including these results. Thanks for addressing my question about the code as well, I apparently did not consult with the Appendix enough. I will update my review and recommend the paper for acceptance.

---

### Official Review · Reviewer_cs6X · 2021-07-16

**Rating:** 7
**Confidence:** 3

**Summary:**

This article study stochastic processes through the lens of kernel methods. The authors propose to use a generalization of kernel mean embeddings to define metrics between stochastic processes. In particular, they used these metrics to define a filtration-sensitive kernel two-sample test. Moreover, they showed how to use this new class of kernels in some learning tasks.

**Limitations And Societal Impact:**

No. The contribution is a general-purpose framework.

**Main Review:**

This work is built upon existing theoretical tools (signature transform, signature kernel). Nevertheless, the use of this kernel seems to be novel in the ML community. From an empirical point of view, the use of this family of kernels seems to improve significantly upon existing methods in the field. However, the evaluation of the kernel is very consuming since it relies on the identity (5), which requires solving a PDE along with some heuristics such as identity (21). It would be interesting to comment in detail on the complexity of such an approximation scheme.

I enjoyed very much reading the paper, and I have some questions:

- In practice, can we go beyond 2nd order MMD with the approximation scheme based on the identity(5)?
-What is the intuition lying behind this transform?
-Is there a toy example where the evaluation of the kernel has an analytical form?

**Time Spent Reviewing:**

7

---

> ### Author Response · Authors · 2021-08-09
> **Official comment for reviewer cs6X**
>
> We thank the reviewer cs6X for the useful feedback. Please find our responses below.
>
> **The  evaluation  of  the  kernel  is  very  consuming  since  it  relies  on  the  identity (5),  which  requires  solving  a  PDE  along  with  some  heuristics  such  as  identity (21).  It would be interesting to comment in detail on the complexity of such anapproximation scheme**
>
> For details about the complexities of the algorithms we kindly refer the reviewer to Appendix B.
>
> **In  practice,  can  we  go  beyond  2nd  order  MMD  with  the  approximation scheme based on the identity (5)?**
>
> Yes, we can iterate the procedure using identities (5) and (21) and we will add the corresponding algorithm in an extra section in Appendix B.
>
> **What is the intuition lying behind this transform?**
>
> We presume that ”this transform” refers to identity (5).  The signature kernel is defined as the inner product between two signatures: $k_S(x,y) =〈S(x),S(y)〉_{H(V)}$.  The fact that this quantity is the solution of the hyperbolic PDE (5), is explained in terms of simple arguments in the proof of Thm. 2.5 in [1]. The sketch of this proof goes as follows: one first shows that this inner product satisfies a double integral equation which comes from the fact that the signature itself solves an integral equation.  Then one uses the fundamental theorem of calculus to differentiate with respect to the two time variables to obtain the PDE.
>
> **Is there a toy example where the evaluation of the kernel has an analytical form?**
>
> To the best of our knowledge the only analytical formula revolving around the signature kernel is given by [2, Equation 1.7].
>
> **References**
>
> [1] Cristopher Salvi, Thomas Cass, Terry Lyons, and Weixin Yang. *The signature kernel is the solution of a Goursat PDE.* arXiv:2006.14794, 2020.
>
> [2] Thomas Cass, Terry Lyons, and Xingcheng Xu. *General signature kernels.* arXiv preprint arXiv:2107.00447, 2021

---

> > ### Comment · Reviewer_cs6X · 2021-09-15
> > **Reviewer reply**
> >
> > Thank you for the response. I suggest adding the pointer to the analytical formula of the toy example in [2].
> >
> > I will keep my original score.

---

### Decision · Program_Chairs · 2021-09-27

**Decision:**

Accept (Poster)

**Comment:**

Our opinion is that this paper should be accepted though there are some weaknesses: it would have been easy in the experiment section to use real-world data from quantitative finance instead of draws from a fractional BM; comparisons to CI tests from the causal inference area; and readability.  On a more detailed level it would be good if the authors would cite a disintegration theorem for their family of measure P_{X|F_{X_t}}.